# Interleukin Networks in GVHD: Mechanistic Crosstalk, Therapeutic Targeting, and Emerging Paradigms

**DOI:** 10.3390/ijms26178620

**Published:** 2025-09-04

**Authors:** Yewei Niu, Chen Liu, Peiyan Li, Jiawei Zhao, Jiamin Jin, Jinfeng Yang

**Affiliations:** 1Department of Immunology, Guilin Medical University, Guilin 541199, China; yewei_niu@stu.glmc.edu.cn (Y.N.); liuchen@stu.glmc.edu.cn (C.L.); lipeiyan@stu.glmc.edu.cn (P.L.); 13256566591@163.com (J.Z.); 2Key Laboratory of Tumor Immunology and Microenvironmental Regulation, Guilin Medical University, Guilin 541199, China

**Keywords:** GVHD, interleukins, allo-HCT, immune regulation, therapeutic targets

## Abstract

Graft-versus-host disease (GVHD) is one of the most prevalent and life-threatening complications that can arise following allogeneic hematopoietic cell transplantation (allo-HCT). GVHD occurs when immune cells—primarily T cells—from the graft recognize host cells as foreign entities and initiate an immune response against host tissues. This immune reaction generally involves a diverse array of cytokines, including interleukins (ILs), which play a pivotal role in modulating the immune response, promoting inflammation, and sustaining immune tolerance. Members of the interleukin family are not only directly involved in the activation, proliferation, and differentiation of T cells but also regulate inflammatory responses and the migration of immune cells. Consequently, they significantly influence both the clinical manifestations and prognosis of GVHD. The objective of this study is to review recent advancements in research concerning interleukins and their role in the pathogenesis of GVHD. This study aims to elucidate how interleukins contribute to immune regulation, inflammatory responses, and clinical manifestations. Furthermore, we will discuss their potential as therapeutic targets, with the intention of providing novel insights and strategies for the clinical management of GVHD.

## 1. Introduction

Allo-HCT represents a potentially curative intervention for a diverse array of conditions, including hematological disorders, metabolic storage diseases, immunodeficiencies, and hematological malignancies. However, acute and chronic GVHD pose significant challenges that must be addressed to enhance the safety of allo-HCT [1]. Acute GVHD primarily affects the skin, gastrointestinal tract, and liver, while chronic GVHD can impact a broader spectrum of tissues and organs. The pathology of GVHD is influenced by various factors released within the inflammatory microenvironment that determine tissue tropism and disease severity [2,3]. The activation of key immune cells involved in this process and the subsequent inflammatory response are often modulated by an array of cytokines—particularly members of the interleukin family. Distinct types of interleukins not only exert direct effects on immune cell functionality but also play a crucial role in determining both the severity of GVHD and associated organ damage by shaping the local immune environment [4]. The graft-versus-leukemia (GVL) effect holds dual significance in the treatment of GVHD [5]. On one hand, it serves as a crucial mechanism for eliminating residual tumor cells following allogeneic hematopoietic stem cell transplantation (allo-HSCT), thereby directly influencing the risk of disease recurrence. On the other hand, GVL and GVHD share certain immune pathways—such as donor T cell activation—creating a challenge in achieving a precise balance during treatment that necessitates both effective control of GVHD and retention of GVL [6].

In the pathogenesis of GVHD, a diverse array of immune cells and their secreted cytokines collaboratively form a complex immune network. Th1 cells, as a significant subset of CD4^+^ T cells, enhance the expression of major histocompatibility complex (MHC) class II molecules and activate immune cells through direct cytotoxicity and the secretion of cytokines such as interferon (IFN)-γ, IL-2, and granulocyte-macrophage colony-stimulating factor (GM-CSF). They play a crucial role in tissue damage and fibrosis in target organs, including the gastrointestinal tract [7,8]. Conversely, Th2 cells exert protective effects primarily by secreting cytokines like IL-4, IL-5, IL-10, and IL-13 [9,10]. These actions include suppressing Th1 responses while promoting M2-type macrophage activation and regulatory T cell (Treg) differentiation; however, they may also contribute to local lesions such as pulmonary fibrosis [11]. Th17 cells are instrumental in mediating tissue inflammation and fibrosis associated with GVHD by releasing key factors such as IL-17, IL-21, and IL-22. Notably, IL-17 facilitates the infiltration of neutrophils and monocytes while activating fibroblasts to produce matrix metalloproteinases along with pro-inflammatory cytokines [10,12,13]. Natural killer (NK) cells exhibit dual functionality: they can protect against GVHD by eliminating activated T cells while secreting anti-inflammatory cytokines like IL-10 and transforming growth factor (TGF)-β; conversely, they may promote GVHD progression through the release of IFN-γ and TNF-α [14,15]. Macrophages dynamically regulate GVHD progression via M1/M2 polarization: M1 macrophages secrete pro-inflammatory factors such as tumor necrosis factor (TNF)-α, IL-1β, and IL-6 that exacerbate tissue damage; meanwhile M2 macrophages secrete anti-inflammatory cytokines like IL-10 and TGFβ that inhibit inflammation while promoting repair processes [16,17]. Dendritic Cells (DCs) serve as pivotal regulators in both the onset and development of GVHD by orchestrating donor T cell activation and differentiation through antigen presentation alongside cytokine secretion coupled with interactions among various immune cell types [18]. Collectively, these immune cells and their intricate cytokine networks establish the immunological foundation for GVHD initiation and progression; however, this study specifically focuses on interleukins to provide mechanistic depth within a defined scope.

Interleukins, commonly referred to as ILs, are a subset of lymphokines that facilitate communication between leukocytes and other immune cells. They belong to the same category of cytokines as hematopoietic growth factors, which coordinate and interact with one another to execute essential hematopoietic and immunomodulatory functions [19]. Interleukins play a crucial role in the transmission of information, activation and regulation of immune cells, mediation of T and B cell activation, proliferation, and differentiation, as well as in the orchestration of inflammatory responses [20].The interleukin family comprises numerous members, with over 40 identified (e.g., IL-1 to IL-40^+^). As interleukins play a pivotal role in the immune system, they have emerged as significant targets for the treatment of various diseases, including rheumatoid arthritis, psoriasis, and cancer [21]. A range of biologics that specifically target interleukins has been developed for clinical use [22,23]. There is a close relationship between interleukins and the pathogenesis of GVHD. In the current study, we summarized the inflammatory roles of interleukins in GVHD pathogenesis and the interactions among different IL-mediated pathways. We also discussed how various mediators exhibited cell type-dependent functions and regulated complex inflammatory cascades. These findings provide mechanistic insights into IL-mediated pathways in GVHD and lay a foundation for developing interleukin-targeted therapies. The contributions of the IL-1 family, IL-2 family, IL-4 family, IL-6 family, IL-10 family, and IL-12 family to GVHD will be discussed in greater detail (Figure 1).

## 2. Interleukins Promoting the Progression of GVHD

In the occurrence and progression of GVHD, pro-inflammatory cytokines play a pivotal role through mechanisms such as the activation of immune cells, disruption of tissue barriers, and promotion of fibrosis [24,25]. Collectively, these pro-inflammatory factors establish a “cytokine storm” network associated with GVHD. The disruption of their dynamic equilibrium directly contributes to tissue damage and disease progression, thereby representing a significant target for therapeutic intervention [26] (Table 1). The roles of each factor will be discussed in detail in the subsequent sections.

### 2.1. IL-1 Family

The first cytokines associated with GVHD-associated tissue lesions are TNF-α and IL-1 [45]. The recruitment and activation of mononuclear effector cells that produce inflammatory cytokines are essential for amplifying the systemic inflammatory response. The IL-1 family members play key roles in GVHD pathogenesis by modulating inflammatory responses and immune cell function [46].

#### IL-1α and IL-1β

IL-1α is mainly produced by epithelial cells, macrophages and B lymphocytes. Unlike IL-1β, which is secreted following processing, IL-1α is generally found intracellularly and is released upon cell death resulting from stress or injury [47]. Caspase-11 has been identified as a factor that exacerbates GVHD by promoting the expansion of donor T cells, facilitating neutrophil infiltration, and inducing intestinal inflammation [29]. The underlying mechanism involves its interaction with lipopolysaccharides (LPSs) and the cleavage of gasdermin D substrates, which subsequently mediates the release of IL-1α. Neutralization of IL-1α significantly mitigated GVHD, indicating that caspase-11 signaling enhances GVHD at least in part through IL-1α. Moreover, it is noteworthy that caspase-11 deficiency does not impact GVL effects, thereby providing new insights for the treatment of GVHD [29]. Studies have demonstrated that prior to and following transplantation, levels of pro-inflammatory cytokines, such as IL-1, are upregulated, leading to a phenomenon known as a “cytokine storm” [48]. In response to this inflammatory surge, the body produces antagonists like IL-1Ra [49]. Polymorphisms in the IL-1Ra gene are correlated with variations in IL-1Ra production levels, which may subsequently influence an individual’s response to IL-1Ra therapy. Animal models and clinical trials have shown the efficacy of IL-1Ra therapy in reducing mortality associated with GVHD; however, its timing and effectiveness can be affected by various factors, including immunosuppressive prophylactic strategies [46]. Furthermore, IL-1 plays a crucial role in the pathophysiology of GVHD while also being indirectly involved in GVL effects [45]. Consequently, polymorphisms within the IL-1 gene—particularly the variable number of tandem repeats (VNTR) polymorphism of IL-1Ra—may be significant for donor selection and could contribute to optimizing GVHD prophylaxis and therapeutic outcomes [30].

IL-1β is mainly produced by activated monocytes, macrophages, and dendritic cells. IL-1β is a distinct molecular form of IL-1 that mediates inflammation not only at the tissue level but also systemically. It plays a crucial role in host defense against infections, as well as in maintaining tissue homeostasis and facilitating repair processes. The first cytokines associated with GVHD-associated tissue lesions are TNF-α and IL-1 [45]. The recruitment and activation of mononuclear effector cells that produce inflammatory cytokines are essential for amplifying the systemic inflammatory response. Initial studies have demonstrated that GVHD is initiated by allogeneic reactive type 1 T cells secreting IFN-γ, which in turn stimulate the production of additional inflammatory cytokines, including TNFα and IL-1 [50]. Furthermore, intestinal commensal bacteria and uric acid play a role in modulating NOD-, LRR- and pyrin domain-containing protein 3 (Nlrp3) inflammasome-mediated IL-1β production [51]. Research has indicated that inhibiting IL-1β or utilizing IL-1 receptor-deficient DCs and T cells enhances survival rates. Notably, early intervention with an IL-1RA or neutralizing anti-IL-1β antibodies administered 24 h prior to transplantation significantly improves survival outcomes in murine models; however, late treatment initiated on day 5 post transplantation proves ineffective. Following HCT, gut commensal bacteria and uric acid contribute to the activation of Nlrp3 inflammasomes, promoting the production of IL-1β [27]. In contrast, gastrointestinal tract interventions involving narrow spectrum antibiotics and depletion of the DAMP uric acid have been shown to mitigate the severity of GVHD [52]. Both Nlrp3 and apoptosis-associated speck-like protein containing a CARD (ASC) are critical for inducing IL-1β synthesis as well as facilitating GVHD progression [53]. In the mouse transplantation model, the microbiota affects DC and T cells by influencing IL-1β production and plays a key pro-inflammatory role in the early stages of GVHD development—particularly associated with Th17 cell response [51]. Clinical data further support these findings [27]. In addition, the irradiated gut microbiota activated the NF-κB pathway via Toll-like receptors (TLR) to drive IL-1β transcription [54]. The gut microbial metabolite N-oxide trimethylamine exacerbates GVHD by inducing the polarization of mouse M1 macrophages and increasing the expression of pro-inflammatory cytokines such as IL-1β, IL-6, and TNF-α [55].Ultimately, these findings underscore IL-1β as a central mediator in GVHD pathogenesis, providing new molecular insights into the pathogenesis underlying autoimmune diseases [28].

### 2.2. IL-2 Family

The members of the IL-2 cytokine family (IL-2, IL-7, IL-9, IL-15, and IL-21) exhibit distinct regulatory effects on GVHD. Specifically, IL-2, IL-9 and their analogs promote the expansion of Tregs, thereby suppressing acute GVHD [56]. Both IL-7 and IL-15 exacerbate GVHD by facilitating the expansion of alloreactive T cells. Meanwhile, IL-21 is crucial for promoting thymic reconstitution but simultaneously drives inflammation mediated by Th17 cells [31]. These cytokines represent critical therapeutic targets for the modulation of GVHD. These research findings offer novel targets and strategies for treating GVHD.

#### 2.2.1. IL-7

IL-7 is a pivotal cytokine in the context of aHSCT, facilitating host extra-thymic T cell expansion [57]. Additionally, IL-7 may enhance the proliferation of allogeneic reactive T cells that are responsible for mediating GVHD. Research has demonstrated that elevated serum levels of IL-7 correlate with the occurrence of acute GVHD [31]. Various cell types have been identified as contributors to IL-7 production, including stromal cells, macrophages, B cells, and thymic epithelial cells. Studies conducted in murine models indicate that radiation exposure during HSCT leads to a reduction in thymic stromal IL-7 production [58]. The expression of IL-7R on T cells is associated with T cell differentiation and maturation based on CD45RA and CCR7 expression profiles [59]. The role of IL-7 in immune reconstitution following HSCT is complex: it promotes thymopoiesis by stimulating the development of immature thymocytes. Some reports suggest that treatment with IL-7 enhances but transiently improves immune reconstitution without increasing allogeneic reactivity [32]. Conversely, other studies indicate that IL-7 may exacerbate GVHD and propose that subsequent blockade of the alpha chain of the IL-7R could mitigate this condition [60].To further complicate matters, IL-7R binds not only through the cell membrane form, but also through the soluble form sIL-7R. Soluble IL-7R binds with similar affinity to membrane-bound IL-7R, and both lead to sIL-7R-mediated inhibition of IL-7 signaling in T cells [61]. The production of sIL-7R occurs not only through the shedding of membrane-bound receptors but is also associated with polymorphisms in the IL-7R gene (rs6897932) [62].

#### 2.2.2. IL-15

IL-15 is a pleiotropic pro-inflammatory cytokine. Research conducted in a mouse model has demonstrated that endogenous IL-15, provided by donor wild-type bone marrow cells, plays a crucial role in the pathogenesis of acute allogeneic GVHD. Donor-derived IL-15 is instrumental in the development of acute GVHD following allogeneic bone marrow transplantation (BMT), as it significantly exacerbates tissue inflammation in both the gut and liver. Furthermore, it contributes to increased morbidity and mortality associated with GVHD by promoting the expansion and activation of allogeneic reactive effector memory CD8^+^ T cells [33]. IL-15 mediated GVHD is entirely dependent on T cells, as the depletion of T cells can completely abolish this process. However, transplantation of IL-15 transgenic bone marrow (BM) cells has been shown to prolong survival in allogeneic hosts compared to wild-type BM cells. This finding suggests for the first time that the level of IL-15 expression in the donor bone marrow may serve as a predictor for clinical outcomes following BMT. Dysregulation of IL-15 after allogeneic BMT results in a shift from a Th2/Tc2 cytokine response to a Th1/Tc1-type response. Importantly, transforming growth TGF-β and other Th2/Tc2 cytokines, such as IL-10, inhibit T cell responses to alloantigens and may also be downregulated in recipients receiving IL-15 transgenic B6 alloantibody BM cells [63]. The cellular response to IL-15 is contingent upon the expression of IL-2Rβ and γc chains, while IL-15Rα functions as a high-affinity receptor that signals through the IL-2/15Rβγc complex. It has been demonstrated that IL-15 is primarily presented to target cells expressing IL-2/15Rβγ in a “trans” manner via IL-15Rα. Emerging evidence indicates that although the expression of IL-15Rα by bone marrow-derived cells is essential for the homeostatic proliferation of splenic memory CD8 T cells, its expression by CD8 T cells themselves is not required [34]. In the experiments, IL-15 transgenic B6 myeloid cells demonstrated a robust capacity to produce and secrete IL-15 protein. This protein is likely captured by IL-15Rα on host antigen-presenting cells (APCs) within lymphoid tissues, facilitating its presentation in trans to donor-derived CD8 T cells [64]. In contrast, non-transgenic IL-15 myeloid cells are incapable of producing cytokines, resulting in less efficient stimulation of donor-derived CD8 T cells by host APCs. This mechanism highlights a unique interaction between donor and host hematopoietic cells mediated through IL-15, suggesting that IL-15 may serve as an important “third signal” for lymphocyte activation during acute graft-versus-host disease (aGVHD).

#### 2.2.3. IL-21

IL-21 is a recently identified member of the common gamma chain family of cytokines [65]. Aurélie Tormo et al. conducted experiments involving BMT and discovered that the administration of IL-21 to transplanted mice significantly enhanced neothymopoiesis by targeting two primary lymphoid tissues: the bone marrow and the thymus. IL-21 was found to inhibit dendritic cell function while simultaneously increasing hematopoietic progenitor cells. More specifically, IL-21 promoted the expansion of short-term hematopoietic stem cells and multipotent progenitor cells within the Lin− Sca-1^+^c-Kit^+^ population in the bone marrow, which serves as a crucial source of building blocks necessary for rapid immune recovery. It is hypothesized that this increase in the pool of bone marrow progenitors enhances cell migration into the thymus [66]. It has been demonstrated that IL-21 directly stimulates T cell development within the thymus by increasing the number of thymic progenitor cells and promoting the recovery of thymic epithelial cells. IL-21 plays a crucial role in immunoglobulin production and facilitates Th17 differentiation in the presence of TGF. However, reports regarding its contribution to Th1 and Th2 differentiation remain limited [35]. Evidence supporting a relationship between IL-21 and autoimmune diseases is accumulating. For instance, overexpression of IL-21 induces inflammation; in the BXSB.6-Yaa^+^/J mouse model of systemic lupus erythematosus (SLE), affected mice exhibit elevated levels of IL-21, whereas IL-21R^−/−^BXSB.6-YAA^+^/J mice do not develop SLE [67]. In a mouse model of GVHD, treatment with an anti-IL-21 neutralizing monoclonal antibody improved survival rates and was associated with reduced production of Th1 cytokines as well as diminished gastrointestinal damage [68].

### 2.3. IL-6 Family

The members of the IL-6 family include IL-6, IL-11, and LIF. Among these, IL-6 plays a pivotal role in the acute inflammatory response. Conversely, both IL-11 and LIF demonstrate protective effects [69,70].

#### IL-6

IL-6 directly influences the differentiation fate of naive CD4^+^ T cells by modulating transcription factors and signaling pathways. Evidence indicates that IL-6 promotes the differentiation of Th17 cells in conjunction with transforming growth factor-β (TGF-β), while simultaneously inhibiting TGF-β-induced Treg differentiation [38]. This bidirectional regulatory mechanism results in an imbalance between Tregs and Th17 cells, which subsequently impacts immune homeostasis. IL-6 collaborates with TGF-β to activate key transcription factors, such as RORγt, thereby driving the differentiation of naive CD4^+^ T cells into Th17 cells. Th17 cells produce IL-17 and other pro-inflammatory cytokines that directly target host tissues, including the intestine and liver, playing a crucial role in autoimmune and inflammatory responses [71]. Furthermore, IL-6 activates downstream signaling pathways—such as JAK/STAT3, MAPK/ERK, and PI3K/AKT—by binding to its receptors (IL-6R and gp130). The phosphorylation of STAT3 is a central event that enhances the transcription of Th17-related genes (e.g., IL-17 and IL-21) while repressing genes associated with Tregs (e.g., Foxp3) [72]. In models of GVHD, blockade of IL-6 signaling—for instance, through anti-IL-6R antibodies—has been shown to mitigate GVHD severity, accompanied by an expansion of Tregs and a reduction in both Th1 and Th17 populations [11]. In humanized mouse models, overexpression of human IL-6 during chronic graft versus host disease (cGVHD) development induced by human Hematopoietic stem and progenitor cells led to distinctive pathological changes in skin, lung and liver tissues, characterized by infiltrating activated macrophages and T cells. The dysregulated IL-6 expression disrupted T cell and myeloid cell homeostasis, synergizing with factors like M-CSF and IFN-α2 to promote macrophage differentiation, TGF-β production, and subsequent tissue fibrosis [36]. Rho-associated coiled-coil containing protein kinase 1(ROCK1) is significantly upregulated in steroid-refractory acute GVHD (SR-aGVHD) patients. Pharmacological inhibition of ROCK1/2 attenuates alloimmune responses by reducing pro-inflammatory cytokine production IL-6 and enhancing IL-10 secretion from donor-derived dendritic cells and macrophages. Mechanistically, ROCK1/2 inhibition downregulates IL-6/JAK-STAT3 signaling and suppresses NF-κB phosphorylation without compromising graft-versus-leukemia effects. Importantly, combination therapy with ROCK1/2 and JAK1/2 inhibitors (e.g., ruxolitinib) demonstrates synergistic therapeutic efficacy, offering a promising treatment strategy for SR-aGVHD [37]. In addition, exosomes derived from MSCs alleviate GVHD-related dry eye by delivering miR-204. miR-204 inhibits the IL-6/IL-6R/Stat3 pathway, converting pro-inflammatory M1 macrophages into anti-inflammatory M2 macrophages and reducing ocular surface inflammation [73].IL-6 drives fibrosis through dual activation of JAK/STAT3 and mTOR pathways. By binding to its receptor, IL-6 phosphorylates STAT3 to upregulate collagen I/III and stimulate fibroblast proliferation [74]. Simultaneously, IL-6 activates mTOR signaling to promote hepatic stellate cell activation and muscle atrophy. These pathways cross-regulate—STAT3 activation induces profibrotic IL-11, which further activates mTORC1, creating a synergistic fibrotic cascade [75].Collectively, these findings underscore the pivotal role of IL-6 as a mediator in both inflammatory and fibrotic processes. Targeting IL-6 may provide novel therapeutic strategies for the treatment of GVHD.

### 2.4. IL-10 Family

Members of the IL-10 family establish a complex “immune-tissue” dialog network in GVHD. IL-26 exacerbates fibrosis and inflammation [76]. IL-10 and IL-22 demonstrates tissue-specific protective and pathogenic effects, while IFN-λ plays a crucial role in maintaining intestinal barrier integrity [77]. Subsequently, we will analyze how the complex interactions among these cytokines influence the progression of GVHD and inform treatment strategies.

#### IL-26

IL-26, a member of the interleukin family, plays a crucial role in pulmonary fibrosis associated with GVHD [78]. Through in vitro studies and mouse model experiments, we have demonstrated that human IL-26 activates murine fibroblasts and enhances collagen production [79]. This effect is particularly pronounced in lung fibrosis related to GVHD-associated occlusive bronchiolitis obliterans. Further investigations revealed that IL-26 is predominantly produced by CD4 T cells infiltrating the lungs, with IL-26^+^ CD26^+^ CD4 T cells significantly contributing to the pathogenesis of occlusive capillary bronchitis. By inhibiting caveolin-1, it is possible to suppress the immune function of donor-derived T cells and reduce IL-26 production, thereby controlling the progression of pulmonary GVHD. Known as a Th17 cytokine, IL-26 can be produced by various cell types including NK cells, macrophages, bronchial epithelial cells, and synovial cells [78]. One study utilizing human IL-26 transgenic (hIL-26Tg) mice and human cord blood mononuclear cells for modeling purposes found that IL-26 significantly increased neutrophil levels in both GVHD target tissues and peripheral blood. Additionally, the expression levels of Th17 cytokines were markedly elevated in donor CD4^+^ T cells derived from hIL-26Tg mice; however, IL-26 did not impact the cytotoxic function of donor CD8^+^ T cells. Furthermore, hIL-26Tg mice exhibited significantly enhanced levels of granulocyte colony-stimulating factor, IL-1β, and IL-6, which exacerbated the systemic symptoms associated with GVHD. These findings not only elucidate a novel role for IL-26 in pulmonary fibrosis related to GVHD but also suggest potential new therapeutic targets for managing clinical chronic GVHD [39].

### 2.5. IL-12 Family

Members of the IL-12 family include IL-12, IL-23, IL-27, IL-35, and IL-39. Among these cytokines, IL-12, IL-23, and IL-39 are known to promote the progression of GVHD. In contrast, IL-35 demonstrates a protective effect against this condition. Notably, IL-27 exhibits both promoting and inhibitory effects in the context of GVHD [80].

#### 2.5.1. IL-12

IL-12 exerts its effects on CD4^+^T cells by activating the STAT3 and STAT4 signaling pathways, which drives naive T cells to differentiate into Th1 cells and significantly enhances IFN-γ production [81]. At the single-cell level, IL-12 not only induces high levels of IFN-γ expression in Th1 cells but also inhibits the production of Th2-type cytokines such as IL-4, thereby establishing a Th1-dominant immune response [41]. This effect is further amplified by a positive feedback loop between IL-12 and IFN-γ: IL-12-induced secretion of IFN-γ enhances the expression of IL-12 receptors, continuously reinforcing signals for Th1 differentiation [82]. Studies conducted in a mouse model of acute GVHD have confirmed that endogenous IL-12 promotes both Th1 differentiation and IFN-γ production, leading to severe immunopathological injury characterized by weight loss and increased mortality rates. Neutralization of IL-12 significantly alleviated GVHD symptoms, suggesting that the differentiation of Th1 cells driven by IL-12—and its associated inflammatory responses involving IFN-γ activated macrophages and cytotoxic T cell attacks on host tissues—constitute critical components in the pathogenesis of GVHD [83]. In a mouse model, IL-12/15/18 pre-activated NK cells exhibited the ability to inhibit aGVHD without compromising GVL activity [84]. This inhibitory effect is closely linked to the sustained expression of Eomes and T-bet, which may also enhance their proliferative potential and long-term survival. IL-12/15/18 preactivated NK cells may influence Treg populations by sequestering IL-2. Studies have demonstrated that NK cells preactivated with IL-12/18 or IL-12/15/18 exhibit memory-like phenotypes in vitro and show enhanced GVL effects in vivo [84,85].

#### 2.5.2. IL-23

Studies have identified a significant role for IL-23, a member of the interleukin family, in the pathogenesis of gastrointestinal graft-versus-host disease (GI-GVHD). It has been demonstrated that elevated levels of IL-23 are associated with an increase in retinoic acid-reactive T cells, particularly CD8 effector T cells exhibiting high expression of RARα, T-bet, and IL-23R at sites of tissue injury in GI-GVHD. These RA-reactive CD8 effector T cells thrive in an IL-23-rich environment and may represent one among several distinct immune cell subsets contributing to the pathogenesis of GI-GVHD [86]. IL-23 serves as a key mediator that links mucosal injury and LPS translocation to subsequent pro-inflammatory cytokine production and GVHD-related pathological damage. The study demonstrates that IL-23 is crucial in mediating colonic GVHD pathology primarily through a T cell-dependent pathway, specifically by inducing IFN-γ secretion, rather than via the Th17 cell pathway. Furthermore, this research confirms the direct involvement of donor APCs in the pathogenesis of GVHD, highlighting IL-23 secretion as a major mechanism underlying disease transmission [42]. IL-23 drives chronic GVHD pathogenesis by promoting pathogenic IFN-γ/IL-17 double-positive Th17 cells, while sparing protective IL-17 single-positive Th17 cells. Targeting IL-23 signaling (via anti-p40 antibodies like ustekinumab) selectively inhibits these pathogenic Th17 populations, offering a promising therapeutic strategy for chronic GVHD [87]. Meanwhile, IL-23 inhibitors selectively suppress pathogenic Th1/Th17 responses while preserving Treg function, offering complementary mechanisms to JAK inhibitors. By reducing inflammatory pressure, IL-23 blockade (e.g., ustekinumab) may indirectly protect Treg cells, making it a synergistic partner for JAK inhibitors in Th17-driven diseases [88]. The findings indicate that broad-spectrum antibiotics, such as imipenem-cilastatin, may cause damage to gut microbiota and exacerbate GVHD severity. This exacerbation may be linked to increased expression of IL-23 by colon myeloid cells, which could subsequently influence the activity of colonic infiltrating CD4^+^ T cells. Therefore, strategically selecting antibiotics to protect gut microbiota and understanding their subsequent effects on interleukin-mediated immune responses may represent a key approach for mitigating the risk of GVHD [89]. Of particular note is the role of the IL-23p19/IL-17 axis in the pathogenesis of aGVHD induced by transplantation of allogeneic cells. It was observed that IL-17 gene expression and serum levels were significantly elevated in BALB/c mice transplanted with WT donor cells. In contrast, deletion of IL-23p19 (p19À/À cells) was associated with increased survival rates and reduced severity of inflammation in these mice. The findings indicate that IL-23p19 may influence aGVHD through both IL-17-dependent and non-dependent pathways [43]. These observations further highlight the complexity inherent in immune regulatory mechanisms involved in GVHD development.

#### 2.5.3. IL-39

IL-39 expression has been found to be elevated in both mouse and human cGVHD, with its overexpression exacerbating the progression of cGVHD. IL-39 plays a crucial role in the pathophysiology of cGVHD by promoting pro-inflammatory responses in T cells, including Th2 cells, and B cells. This effect may occur through interaction with the IL-39 receptor on T cells and subsequent activation of the STAT pathway [44]. Additionally, IL-39 influences T cell selection within the thymus, regulates the frequency of Treg cells, and may be produced by activated B cells, CD8^+^ T cells, and CD11b^+^ myeloid cells [90]. Furthermore, IL-39 is highly expressed in the gastrointestinal tract of cGVHD mice, which correlates strongly with disease severity. Thus, it may serve as a potential prognostic and diagnostic biomarker for patients suffering from cGVHD [44].

## 3. Interleukins Inhibit the Progression of GVHD

During the pathological process of GVHD, various anti-inflammatory factors play protective roles by precisely modulating immune responses (Table 2). Among the IL-2 family, IL-2 significantly inhibits the development of acute GVHD by enhancing Treg populations, and its analogs have demonstrated clinical applicability in treatment [91]. However, certain cytokines within the traditional anti-inflammatory interleukin family exhibit opposing effects. This article will elaborate on the roles of each factor in subsequent sections.

### 3.1. IL-2 Family

#### 3.1.1. IL-2

Orthogonal IL-2 (oIL-2), a variant of IL-2 engineered through protein engineering, has been utilized along with other IL-2 variants to establish an orthogonal IL-2/IL-2Rβ system. By employing engineered oIL-2Rβ Tregs in conjunction with oIL-2, the researchers successfully achieved selective expansion of Tregs in vivo, while circumventing the activation of allogeneic reactive T cells [106]. This innovative approach not only significantly enhanced the protective effects of Tregs on target organs affected by aGVHD, but also effectively inhibited the proliferation of donor T cells and the production of inflammatory cytokines. Consequently, this led to a reduction in aGVHD severity and improved survival rates. Moreover, this strategy demonstrated no adverse off-target effects while preserving the GVL, thereby offering a promising new therapeutic avenue for enhancing the prognosis of recipients undergoing allo-HSCT [93].Blocking the binding of IL-2 to the IL-2 receptor on T cells through the administration of anti-IL-2 monoclonal antibodies (e.g., JES6) has proven effective in preventing GVHD while preserving robust GVL activity. This protective effect is contingent upon the expression of Programmed Death-Ligand 1 (PD-L1) in host tissues, which inhibits pathogenic T cell function by obstructing the activation of the IL-2-Stat5 signaling pathway in donor T cells. Consequently, this leads to a reduction in GM-CSF production and promotes T cell incompetence, depletion, or differentiation into IL-10 producing Tr1 cells. Furthermore, JES6 treatment capitalizes on variations in PD-L1 expression between GVHD target tissues and lymphoid tissues to selectively maintain GVL activity within lymphohematopoietic compartments while simultaneously preventing GVHD in parenchymal tissues. These findings provide a compelling rationale for developing novel therapeutic strategies aimed at targeting both the IL-2 and PD-1/PD-L1 pathways without compromising GVL activity as a means to prevent GVHD [92].Through advanced genetic code expansion techniques, researchers have successfully developed IL-2 variants that covalently bind to IL-2Rα, including L72-FSY and its polyethylene glycol-conjugated form, PEG-L72FSY. These variants preferentially activate and expand Tregs in a highly specific and sustained manner by enhancing their binding affinity and recirculation while simultaneously reducing the activation of effector T cells (Teffs). In in vivo experiments, both L72-FSY and PEG-L72FSY demonstrated therapeutic superiority over wild-type IL-2, effectively alleviating symptoms associated with GVHD, SLE, and other inflammatory conditions. These findings offer promising avenues for the development of clinically relevant IL-2 therapies and may facilitate the creation of mutant forms of other key cytokines, potentially leading to significant improvements in their therapeutic efficacy through modulation of cellular transport [107].

#### 3.1.2. IL-9

High levels of IL-9, a crucial factor for mast cell growth and activation, are produced by activated Treg. The data indicate that IL-9 serves as a functional link between the recruitment of activated Treg cells and the activation of mast cell-mediated regional immunosuppression; notably, neutralization of IL-9 markedly accelerates allograft rejection in tolerant mice. These findings establish IL-9 as a pivotal immunosuppressive mediator in the Treg-mast cell axis, highlighting this cytokine as a potential therapeutic target for modulating immune responses [19]. A new subpopulation of T cells identified in recent studies, Th9, is mainly characterized by the secretion of IL-9, and Th9 cells develop from CD4 precursor cells stimulated by TGF-β and IL-4 [108]. In the context of transplantation, Th9 cells demonstrate tolerance, and fractionated IL-9 has been shown to promote allograft tolerance [109]. Rapamycin is an immunosuppressant that inhibits cell signaling by blocking mTOR. Research indicates that CD4^+^ T cells can differentiate into Th9 cells in response to co-stimulation with TGF-β and IL-4, regardless of the presence or absence of rapamycin. These differentiated Th9 cells secrete IL-9 and exhibit a transcription factor profile characteristic of both Th9 and Th2 lineages (high GATA-3/low T-bet). Rapamycin-resistant Th9 cells significantly reduced the implantation of CD4^+^ and CD8^+^ T cells while inhibiting IFN-γ secretion in allogeneic grafts. Notably, these Th9 cells displayed high phenotypic stability in vivo with limited differentiation plasticity, maintaining elevated levels of IL-9 expression even in the presence of rapamycin while effectively inhibiting IFN-γ-driven allogeneic responses. This study represents the first demonstration of the phenotypic stability exhibited by Th9.R cells during allogeneic (BMT), revealing their capacity to inhibit type I cytokine production in allogeneically reactive T cells while preserving their ability to secrete IL-9 [110]. Meanwhile, Ramadan’s study revealed that IL-33 enhances the expression of mST2, IL-9, and PU.1 in IL-9-expressing CD4 (Th9) and CD8 (Tc9) cells. The expression of PU.1 is mediated through its binding to the stimulation expressed gene 2 (ST2) promoter as well as the IL-9 gene promoter, thereby regulating IL-9 production while inhibiting Th2 cytokine secretion. The researchers demonstrated that IL-9-producing T cells activated via the ST2-IL-33 pathway (referred to as T9_IL-33_ cells) were capable of enhancing GVL activity while simultaneously reducing GVHD. This was achieved through two opposing mechanisms observed in both murine models and human subjects: firstly, by expressing biorhythmic proteins that protect the host from lethal immune responses; secondly, by upregulating CD8α expression to bolster antileukemic activity—this effect being superior compared to T9, T1, or untreated T cells. Consequently, the over-transfer of allogeneic T9_IL-33_ cells presents a promising therapeutic strategy for achieving GVL activity while mitigating GVHD [94].

### 3.2. IL-4 Family

IL-4 and IL-13, as Th2-type cytokines, play complex and nuanced regulatory roles in the context of GVHD [111]. IL-4 contributes to the pathogenesis of GVHD through a dual mechanism, whereas IL-13 primarily exerts its protective effects via the JAK/STAT6 signaling pathway [95,112]. These findings not only enhance our understanding of GVHD’s underlying mechanisms but also identify critical targets for the development of novel immunomodulatory therapies.

#### IL-13

IL-13 is a cytokine produced by activated Th2 cells, mast cells, and basophils, belonging to the IL-4/IL-13 family. Notably, IL-13 can inhibit the expression of pro-inflammatory cytokines such as IL-1β, TNF-α, and IL-6 in monocytes and macrophages, which contributes to reducing tissue damage [98]. This inhibitory effect plays a crucial role in alleviating the inflammatory response associated with GVHD and mitigating damage to host tissues. Previous studies have demonstrated that infusion of donor-derived innate Type 2 innate lymphoid cells (ILC2) can prevent and treat aGVHD in the lower digestive tract without compromising GVL effects. However, this clinical translation method presents challenges. The functionality of ILC2 cells is contingent upon their expression of both IL-13 and AREG. The ability to generate third-party ILC2 cells thus represents an innovative approach toward preventing aGVHD [113]. IL-13 can stimulate the generation of a specific subpopulation of myeloid-derived suppressor cells (MDSC)-IL-13, which significantly inhibits allogeneic T cell responses by upregulating arginase-1 expression. The inhibitory effect of MDSC-IL-13 on allogeneic T cell responses is markedly more effective than that of conventional MDSCs. Although both MDSCs and MDSC-IL-13 can mitigate the lethality associated with GVHD, it is evident that MDSC-IL-13 exhibits superior efficacy. Notably, MDSC-IL-13 does not compromise the graft-versus-leukemia effect exerted by donor T cells. These findings reveal a novel strategy for preventing GVHD through the application of MDSC-IL-13 and pegylated human arginase-1 [95]. Although some studies have correlated IL-13 levels with the severity of GVHD in patients [114], experiments conducted in established mouse models of GVHD revealed that transplantation with IL-13^−/−^ cells resulted in increased mortality and decreased Th2 cytokine levels, alongside elevated serum TNF-α levels. Further investigations demonstrated that IL-13 inhibits TNF-α production following allogeneic bone marrow transplantation while enhancing the secretion of IL-4 and IL-5, thereby supporting the notion that IL-13 plays a protective role in GVHD [97]. Ultimately, a study examining the combined deficiency of IL-4, IL-5, IL-9, and IL-13 within T cells using a mature mouse model indicated that this collective absence of Th2 cytokines led to enhanced T cell proliferation as well as elevated serum concentrations of TNF-α, IL-2, IFN-γ, and IL17a—culminating in exacerbated manifestations of GVHD [115].

### 3.3. IL-6 Family

#### 3.3.1. IL-11

IL-11 is a member of the IL-6 cytokine family. Initially identified as a hematopoietic cytokine, IL-11 has since been shown to exert effects in various tissues, including those of the immune system, gastrointestinal tract, liver, nervous system, as well as bone and adipose tissue [116]. Research indicates that IL-11 enhances platelet production following bone marrow suppression therapy and mitigates gastrointestinal damage [117,118]. Additionally, it exhibits potent anti-inflammatory properties. Recent studies have demonstrated that IL-11 significantly inhibits GVHD in well-characterized mouse models against MHC and minor antigens [100]. Specifically, IL-11 reduces GVHD-related damage to the small intestine during radiation conditioning by preventing the translocation of LPSs from the intestinal lumen into systemic circulation. Furthermore, IL-11 can inhibit the increased secretion of TNF-α and IL-12 by host macrophages. Consequently, during donor T cell activation, IL-11 promotes polarization towards a type 2 cytokine response. Simultaneously, it regulates multiple steps within the cytokine cascade associated with GVHD. These attributes position IL-11 as a promising candidate for adjuvant therapy aimed at preventing GVHD [119]. Studies have demonstrated that IL-11 inhibits GVHD by suppressing the production of inflammatory cytokines within the cytokine cascade during acute GVHD, while simultaneously preserving Cytotoxic T lymphocyte (CTL) function. Research indicates that IL-11 treatment reduces intestinal damage caused by GVHD, prevents LPS translocation, and inhibits the heightened secretion of TNF-α and IL-12 by host macrophages. The selective inhibition of inflammatory cytokines by IL-11, combined with the preservation of T cell-mediated tumor lysis mechanisms, offers a novel strategy for differentiating between GVHD and GVL effects. This approach may serve as an effective adjunct in clinical protocols aimed at preventing GVHD [100]. Current immunosuppressive agents such as cyclosporine A and prednisone provide substantial but incomplete protection against GVHD; moreover, these agents can adversely affect GVL effects by dampening T cell activity [120,121]. Therefore, short-term administration of IL-11 may represent a promising new strategy to separate GVHD from GVL effects and could act as an effective adjuvant in clinical protocols for preventing GVHD.

#### 3.3.2. LIF

Leukemia inhibitory factor (LIF), a pleiotropic cytokine belonging to the IL-6 superfamily, was initially identified as a factor that induces differentiation in myeloid leukemia cells while simultaneously inhibiting their proliferation. Subsequent studies have demonstrated that LIF exhibits diverse biological functions across various physiological and pathological contexts, with effects that are highly dependent on cell type, tissue microenvironment, and specific biological circumstances. Emerging evidence indicates that LIF plays critical roles within stem cell niches, where it maintains homeostasis and promotes regeneration in multiple somatic tissues including the intestine, nervous system, and muscle [122]. Furthermore, LIF acts as an important immunomodulator and demonstrates protective effects in various immunopathological conditions such as infections, inflammatory bowel disease (IBD), multiple sclerosis and GVHD [102]. The study conducted by Jianming Wang et al. revealed that administration of recombinant leukemia inhibitory factor (rLIF) protected mice from GVHD-induced tissue damage and mortality without compromising graft-versus-leukemia activity. rLIF treatment resulted in reduced infiltration and activation of donor immune cells while preserving intestinal stem cell function, thereby ameliorating GVHD symptoms. Mechanistically, rLIF activates the STAT1 signaling pathway to downregulate IL-12-p40 expression in irradiated recipient dendritic cells; this leads to decreased MHC class II expression on intestinal epithelial cells and consequently reduces donor T cell activation and infiltration. These findings underscore the crucial role of LIF in preventing GVHD through modulation of the STAT1/IL-12/MHC-II axis while supporting intestinal stem cell function and suppressing donor T cell infiltration and activation [101]. The immunoregulatory properties of LIF position it as a promising therapeutic candidate for a range of immune-related disorders, including viral and bacterial infections, as well as IBD. In contrast, small-molecule inhibitors that target the LIF/LIFR signaling pathway have demonstrated antitumor efficacy in murine models. Given the pleiotropic roles of LIF in both physiological processes and pathological conditions—particularly concerning immune disorders and cancer—further elucidation of its mechanisms of action will enhance the development of novel therapeutic strategies [123].

### 3.4. IL-10 Family

#### IFN-γ

The IFN-γ receptor is a unique heterodimer composed of the IFN-λ and IL-10RB chains [124]. In humans, four ligands are expressed: IFNL1 (IL-29), IFNL2 (IL-28A), IFNL3 (IL-28B), and IFNL4, while mice express only IFNL2 and IFNL3 [125]. Henden et al. demonstrated that IFN-λ mitigates the loss of intestinal stem cells and promotes the regeneration of intestinal epithelium, thereby preserving mucosal barrier function in a murine model of aGVHD. This study identified interferon λ as a critical protective factor in the immunopathology of gastrointestinal GVHD, particularly highlighting its significant role within the ISC compartment. Ifnlr1^−/−^ mice exhibited exacerbated gastrointestinal GVHD and increased mortality; this effect was independent of alterations in PAN cells or microbiome composition. The growth of intestinal organoids from Ifnlr1^−/−^ mice was significantly impaired, indicating that targeting Ifnlr1 deficiency had intrinsic effects on Lgr5^+^ ISCs and NK cells. Mice treated with polyethylene glycolated recombinant IL-29 (PEG-rIL-29) displayed an increased number of Lgr5^+^ ISCs along with enhanced organoid growth; notably, this effect occurred independently of IL-22 or type I interferon signaling while regulating gene sets associated with proliferation and apoptosis in Lgr5^+^ ISCs. Furthermore, PEG-rIL-29 treatment improved survival rates following BMT, reduced the severity of GVHD, and promoted epithelial proliferation alongside ISC-derived organoid growth. After BMT, PEG-rIL-29’s protective effect on ISC numbers was observed regardless of whether recipient NK cells were exposed to IFN-λ signaling. Thus, targeting IFN-λ presents an attractive strategy for preventing ISC loss and mitigating immunopathological damage during GVHD [103].

### 3.5. IL-12 Family

#### IL-35

IL-35, a member of the interleukin family composed of p35 and EBI3, serves as a crucial regulatory cytokine primarily secreted by CD4^+^ Foxp3^+^ Tregs and Bregs [126]. It plays a significant role in suppressing inflammation and mitigating the severity of autoimmune diseases [127]. Evidence indicates that IL-35 exhibits inhibitory effects in aGVHD mouse models, while serum levels of IL-35 are markedly diminished in patients with high-grade GVHD. Furthermore, IL-35 has been linked to an increased frequency of Tregs, decreased Th1 differentiation, and a reduction in GVHD incidence [128]; it also appears to sustain GVL activity. However, within the tumor microenvironment, IL-35 may counteract the T cell responses necessary for effective GVL effects by inducing effector cell depletion. Consequently, despite its evident immunomodulatory properties that may aid in preventing GVHD, comprehensive investigations into its specific impacts on T cell anti-tumor responses are essential prior to clinical application [104].

### 3.6. IL-17 Family

Members of the IL-7 family include IL-17 and IL-25 [129]. Among these, the IL-17 cytokine family exhibits a dual regulatory role in the pathogenesis of GVHD [130]. IL-25 mitigates GVHD by decreasing plasma levels of interferon-γ and interleukin-6, along with other mechanisms [131]. These findings underscore the complex roles played by members of the IL-17 family in both protective and pathogenic processes during GVHD.

#### IL-25

In the pathogenesis of GVHD, the gastrointestinal tract frequently emerges as the primary organ affected, with the extent of damage closely correlating to the severity of the condition [132]. Several studies have investigated the role of cup cells in gastrointestinal GVHD. Intestinal cuprocytes contribute to forming a mucus layer that serves as a barrier between intestinal microbiota and host tissues. The loss of cup cells is recognized as one of the histological hallmarks of GVHD, resulting in disruption of this protective mucus layer and an increase in bacterial translocation [105]. Pre-transplant administration of IL-25 has been shown to safeguard cuprocytes from GVHD, mitigate bacterial translocation, lower plasma levels of IFN-γ and IL-6, and alleviate symptoms associated with GVHD. The protective effect exerted by IL-25 is contingent upon Lypd8, an antimicrobial molecule derived from intestinal epithelial cells that inhibits motility in flagellated bacteria [105].

## 4. Interleukins with Dual Effects on GVHD Progression

Interleukins play a complex and bidirectional regulatory role in the progression of GVHD. Their functions within this condition are intricate and nuanced, defying simple categorization into a single aspect. The dynamic network established by these cytokines is crucial in determining the pathological processes associated with GVHD, thereby offering significant opportunities for targeted therapeutic interventions (Table 3).

### 4.1. IL-1 Family

#### 4.1.1. IL-33

The interleukin family, particularly IL-33 and its receptor growth ST2, plays a crucial role in the pathogenesis of GVHD. IL-33 enhances type 1 immune responses and drives immune regulation; its activity is influenced by the expression of ST2 on cells and the presence of pro-inflammatory cytokines. Therapeutic administration of IL-33 during allo-HCT expands Tregs, particularly ST2^+^ Treg subsets, which are essential for preventing acute GVHD [146]. IL-33 exerts its effects on GVHD by inhibiting M1-type macrophage activation while promoting the function of granulocyte MDSCs, thereby providing protective effects against GVHD. Conversely, loss of Tregs results in an increase in harmful M1-type macrophages that activate donor effector T cells and mediate infiltration into target tissues affected by GVHD. By carefully orchestrating the expansion of ST2^+^ Tregs, it may be possible to counteract deleterious type 1 alloimmune responses mediated by IL-33, thus limiting the incidence of GVHD [134]. Research has indicated that ST2^+^ Tregs are predominantly localized in the intestinal tract, which is the primary target organ affected by GVHD [147]. IL-33 promotes the expansion of ST2^+^ Tregs, enhances their immunosuppressive function, and facilitates tissue repair by upregulating bi-regulatory proteins. This dual mechanism—immunosuppression coupled with tissue protection—effectively mitigates the severity of acute GVHD. Consequently, employing IL-33-stimulated Tregs as a form of cell therapy presents a novel therapeutic strategy for preventing GVHD in non-malignant contexts [135]. However, the role of IL-33 in immune responses is complex and may vary depending on the specific disease model, cell type involved, and inflammatory context. During GVHD progression, the presence of IL-23 may counteract the tolerogenic effects of IL-33 on Tregs without influencing IL-33-induced activation of effector T cells. Moreover, both IL-33 and IL-18 have been found to synergistically enhance the inflammatory potential within human Th1 and Th2 cultures [148]. Conversely, allogeneic reactive T cells that lack signaling from IL-33 produce lower levels of IFN-γ and exhibit reduced pathogenicity while still retaining cytotoxic activity essential for graft-versus-leukemia responses. Inhibition of the interaction between IL-33 and ST2 has demonstrated efficacy in reducing pro-inflammatory cytokine production, mitigating tissue damage, and decreasing GVHD-related mortality [136,149]. Therefore, modulation of the IL-33/ST2 axis represents a significant target for preventing GVHD; however, further research is necessary to elucidate underlying mechanisms and identify optimal intervention strategies [133,150].

#### 4.1.2. IL-18

The interleukin family, particularly IL-18, plays a crucial role in acute myeloid leukemia (AML) and its associated complications, such as GVHD. Research has demonstrated that higher expression levels of IL-6 and IL-18 have been observed in patients who develop aGVHD following bone marrow transplantation [151]. While IL-18 has been shown to induce IFN-γ production and CTL activity under specific conditions both in vitro and in vivo, neutralization of IL-18 did not inhibit IFN-γ production or CTL activity in an experimental model of acute GVHD. This suggests that IL-18 may not be essential for T cell activation during acute GVHD [152]. In immunodeficient mice implanted with human peripheral blood mononuclear cells (PBMC), administration of IL-18 resulted in an increase in the population of CD8^+^ T cells while significantly reducing the number of CD4^+^ CD25^+^ FoxP3^+^ Tregs. This alteration in the ratio of effector T cells to Tregs accelerated the onset of heterozygous GVHD and exacerbated its severity. Furthermore, IL-18 has shown potential in cancer treatment. It may help restore the immune function of cancer patients by suppressing the activity of Treg cells and promoting the function of conventional T cells [137]. On the contrary, administration of IL-18 can effectively prevent the occurrence of chronic GVHD without causing the immune deficiency and mortality associated with acute GVHD. IL-18 exerts its therapeutic effects on chronic GVHD through a tripartite mechanism: it induces donor-versus-host CD8^+^ CTLs, down-regulates MHC class II expression in host B cells, and reduces the population of donor CD4^+^ T cells [138]. In studies utilizing mouse models, NK cells pre-activated with IL-12/15/18 demonstrated the ability to sustain Eomes and T-bet expression while effectively suppressing acute GVHD without compromising GVL effects. Moreover, IL-12/15/18 pre-activated NK cells were unable to significantly extend survival in hormonal mice; this may be attributed to the involvement of IFN-γ as well as variations across tumor models. Interestingly, the upregulation of CD25 on these NK cells may have enabled them to competitively sequester IL-2, thereby influencing Treg function without leading to an exacerbation of acute GVHD severity [85]. These findings not only offer a novel strategy for treating chronic GVHD in murine models but also provide renewed hope for addressing human autoimmune diseases.

### 4.2. IL-4 Family

#### IL-4

IL-4 exhibits complex regulatory effects in GVHD, demonstrating distinct spatiotemporal specificity. As a classic type 2 cytokine, IL-4 antagonizes the functions of IFN-γ and IL-12, while also inhibiting CTL differentiation [153]. This suggests that IL-4 should theoretically suppress acute GVHD while promoting the development of chronic GVHD. Clinical observations support this hypothesis: among cancer patients undergoing allogeneic bone marrow transplantation, those exhibiting stronger IL-4 responses experience significantly less severe GVHD compared to individuals with weaker responses [154]. Furthermore, animal studies reveal that IL-4 derived from basophils, Natural killer T (NKT) cells, and conventional CD4^+^ T cells can effectively inhibit GVHD. Additionally, exogenous administration of IL-4 demonstrates alleviating effects on acute GVHD [139].The Charles S. Via team conducted a systematic investigation into the mechanism of IL-4 action within a non-irradiated parent-to-F1 GVHD model, which simulates kidney transplant rejection. Their research revealed that the simultaneous administration of IL-4 with donor cells completely inhibited the generation of CD8^+^ CTLs. However, delaying this administration by three days negated its protective effect [154]. Importantly, IL-4 did not influence the homing of donor CD8^+^ T cells to the spleen; rather, it specifically suppressed their differentiation into CTLs [155]. Utilizing models deficient in IL-4Rα, they confirmed that IL-4 primarily exerts its protective effects through regulation of host cells instead of acting directly on donor cells. Mechanistically, IL-4 reduces specific subsets of splenic dendritic cells—thereby diminishing antigen presentation—while simultaneously increasing both neutrophil and CD8^+^ T cell populations, thus enhancing an immunosuppressive microenvironment. Interestingly, the immune responses induced by helminth infections, characterized by a Th2-type profile, offer new insights into the dual role of IL-4. Helminth stimulation enhances IL-4 secretion, which in turn promotes the expansion of Foxp3^+^ Tregs through a TGFβ-dependent pathway, thereby exerting a suppressive effect on GVHD. This process is contingent upon the production of IL-4 derived from host cells and the expression of the transcription factor GATA binding protein 3 (GATA3), ultimately establishing a TGFβ-Treg immunoregulatory axis [156]. Regarding the cellular sources of IL-4, Ji Hyung Kim et al. utilized knockout models to demonstrate that type II NKT cells (non-Vα14Jα18 type) in the bone marrow play a crucial role in providing protection against GVHD. Mice receiving donor bone marrow deficient in IL-4 (B6.IL-4^−/−^ or Jα18^−/−^IL-4^−/−^) exhibited significantly reduced levels of IL-4 and IL-10 production in splenocytes compared to wild-type or IFN-γ-deficient groups. This finding confirms that type II NKT cells mediate protective effects through an IL-4-driven Th2 immune deviation [157]. Together, these studies elucidate the pleiotropic effects of IL-4 in GVHD. During the acute phase, IL-4 mitigates pathological damage by modulating the host’s immune microenvironment. Conversely, in chronic phases, it may contribute to dysregulated immune responses. Various cellular sources of IL-4—such as NKT cells and CD8^+^ T cells—engage in disease progression through distinct mechanisms, thereby offering precise therapeutic targets for clinical intervention.

### 4.3. IL-10 Family

#### 4.3.1. IL-10

A critical process in the pathogenesis of aGVHD is the activation of CD8^+^ effector T cells by DCs. Inhibition of ROCK1/2 has been shown to reduce CD8^+^ cell proliferation by targeting DCs, while simultaneously increasing the expression of IL-10 mRNA, which plays a significant anti-inflammatory role during aGVHD. Furthermore, knockdown of ROCK1 in macrophage cell lines also led to decreased production of inflammatory cytokines and an upregulation of IL-10 production [37]. These findings are consistent with studies indicating that the transfer of M2 macrophages into GVHD mice can ameliorate aGVHD [158]. However, in the studies conducted by Sojan Abraham et al., IL-10 exhibited complex and diverse effects on IL-2-induced T cell expansion within the NSG mouse model of human GVHD. When administered concurrently with IL-2, IL-10 initially inhibited the activation and expansion of responding T cells. However, it subsequently modulated the immune response to induce a substantial oligoclonal expansion of CD4^+^ T cells, ultimately resulting in GVHD lethality [140]. Furthermore, blockade of the IL-10 receptor has been shown to reverse the pathogenic effects of IL-10 in vivo, indicating that inhibiting IL-10 signaling may offer a potential therapeutic strategy for treating GVHD [159].

#### 4.3.2. IL-22

IL-22 is a member of the IL-10 cytokine family [160]. IL-22 is primarily produced by lymphocytes including Th1 cells, Th17 cells, Th22 cells, CD8^+^ T cells, and ILC3 cells [161]. Research conducted by Teshima’s group demonstrated that ISCs are responsible for regenerating intestinal epithelium following injury; however GVHD significantly impairs ISC recovery capabilities. ISCs express receptors for IL-22; studies have indicated that defects in these receptors lead to increased tissue damage and mortality rates. In the gut environment specifically, production of IL-22 predominantly occurs through receptor-derived ILCs. A deficiency in the IL-22 receptor results in heightened crypt cell apoptosis along with ISC depletion and loss of epithelial integrity [162]. IL-22, produced by intestinal ILC3 cells, facilitates the regeneration of intestinal epithelium and enhances barrier function [163]. In preclinical models of acute GVHD, treatment with recombinant IL-22 has been shown to alleviate disease symptoms, increase the population of Lgr5^+^ stem cells, upregulate the expression of antimicrobial peptides such as Reg3α and Reg3γ, and improve epithelial integrity. The beneficial effects of IL-22 extend beyond the gastrointestinal tract; it is also essential for the regeneration and survival of thymic epithelial cells. Specifically, IL-22 derived from thymus-resident ILC3 supports thymic regeneration during allogeneic transplantation [164]. However, during intestinal inflammation, the overproduction of IL-22 in conjunction with other pro-inflammatory cytokines can create a pro-inflammatory cytokine environment. Supporting this perspective, a recent study by Bachmann et al. reported a synergistic effect between IFN-α and IL-22 that resulted in the activation of STAT1 in a human colon cell line. Studies observed a synergistic effect between IL-22 and IFN-α in isolated mice treated with colonic explants, resulting in an enhancement of the STAT1/CXCL10 signaling pathway. These findings suggest that IL-22 may facilitate Th1 cell infiltration—the predominant pathological T cell subset involved in acute GVHD—via STAT1 and CXCL10 within an IFN-α-rich environment [165]. The levels of CXCL10 were found to increase following exposure to a combination of IL-22 and type I IFN effects. CXCL10 serves to recruit effector T cells expressing CXCR3 to sites of tissue injury; furthermore, inhibition of CXCR3 has been shown to mitigate the severity of GVHD in murine models [166]. Deletion of IL-22 in donor cells and the blockade of the type I interferon signaling pathway in recipient mice result in a reduction in Th1 inflammation within the recipient gut. Recent studies have demonstrated that IL-22 enhances neutrophil accumulation in peripheral tissues, particularly in the liver and lungs, through its role in producing major neutrophil chemotactic factor CXCL1 [143]. Similarly, neutrophils recruited to intestinal tissues of GVHD recipients contribute to tissue damage and exacerbate intestinal GVHD [167].

### 4.4. IL-12 Family

#### IL-27

The pro-inflammatory and anti-inflammatory effects of IL-27, a member of the IL-12 cytokine family, have garnered significant research attention [168]. Recent studies have demonstrated that blockade of the IL-27 signaling pathway effectively prevents fatal GVHD. This protective effect is primarily attributed to enhanced reconstitution of all Treg subpopulations and increased stability of Foxp3 expression. Moreover, inhibition of IL-27 selectively diminishes IL-10 production in conventional T cells while leaving IL-10 production in Treg cells unaffected. This selective modulation preserves the ability of Tregs to suppress GVHD through this mechanistic pathway [144]. On the other hand, IL-27 shows significant anti-inflammatory effects in a variety of autoimmune diseases. For instance, in experimental autoimmune encephalitis (EAE) and rheumatoid arthritis models, IL-27 alleviates inflammatory responses by inhibiting the differentiation of Th17 cells and the production of IL-17 [169]. In addition, IL-27 may additionally be considered a pre-conditioning factor potentially strengthening Treg suppressive activity when it comes to adoptive Treg transfer therapy [170]. Meanwhile, research has demonstrated that hPMSCs can induce the production of CD4^+^ IL-10^+^ IFN-γ^+^ T cells, which possess immunomodulatory functions. Notably, in inflammatory microenvironments, IL-27 potentiates PDL2 expression via the JAK/STAT signaling pathway to functionally modulate hPMSCs. This regulatory mechanism subsequently enhances the capacity of hPMSCs to induce the generation of CD4^+^IL-10^+^IFN-γ^+^ T cells. These findings provide new insights into understanding the role of interleukin family members in GVHD and highlight potential clinical applications for hPMSCs [145].

### 4.5. IL-17 Family

#### IL-17

Early studies have demonstrated that in vitro cytokine polarization of Th17 cells can induce GVHD, particularly affecting the skin and lungs [171]. Recent investigations have revealed that receptors incapable of producing or signaling IL-17 lead to hyperacute intestinal GVHD. When WT mice were co-housed with IL-17RA and IL-17R-deficient mice, it was observed that WT mice exhibited increased severity of GVHD, accompanied by a shift in gut microbiota towards that found in IL-17RA/C mice [172]. It has been established that IL-17A-producing Th17 cells are instrumental in inducing GVHD [173], while IL-17A cytokines confer a protective effect following allo-BMT [130]. Notably, CD4 T cells deficient in IL-17A showed a diminished capacity to induce GVHD; this reduction was correlated with lower levels of IFN-γ production by CD4 T cells, macrophages, and granulocytes during the early phase post BMT [174]. Conversely, when splenocytes or T cells from IL-17A^−/−^ donors were transferred into sublethally irradiated recipients, the absence of IL-17A resulted in heightened lethality associated with GVHD and an increase in Th1 cell differentiation within the recipients [173]. Furthermore, it was noted that IL-17 could downregulate the Th1 response by inhibiting donor macrophage production of IL-12; depletion of macrophages in vivo diminished the protective effects attributed to IL-17 [175]. The protective function of IL-17A within the context of GVHD may stem from its dual role played by Th17 cells — exhibiting both protective and pathogenic characteristics. In allo-BMT of different models, the differences in cytokine environments caused by different radiation intensities and different GVHD development dynamics may affect whether Th17 cells and IL-17A exert pro-inflammatory or anti-inflammatory effects in GVHD [130]. It is noteworthy that protective subsets secrete anti-inflammatory cytokines such as IL-10 and TGF-β. In contrast, pathogenic Th17 cells primarily produce pro-inflammatory mediators including IFN-γ and GM-CSF [176].

## 5. Role of Interleukins in Clinical Trials of GVHD

Animal studies have revealed a complex interleukin network governing GVHD pathogenesis, offering novel therapeutic approaches like ‘metabolism-immune co-regulation’. However, key limitations persist: human immune heterogeneity and transplant variables create additional regulatory complexity beyond animal models, and the clinical relevance of several interleukins remains unclear. The following discussion will evaluate recent clinical evidence on interleukin-targeted strategies to inform future personalized therapies (Table 4).

IL-15 and IL-7 levels serve as predictive biomarkers for acute GVHD and tumor recurrence post HCT. Thiant et al. findings suggest that early monitoring of IL-7/IL-15 kinetics could guide clinical interventions, with IL-15 pathway inhibition representing a potential therapeutic strategy for acute GVHD prevention [31]. Hippen et al. demonstrated the therapeutic potential of IL-21 pathway inhibition in GVHD prevention through both clinical and preclinical studies. Prophylactic neutralization of IL-21 with monoclonal antibodies significantly improved GVHD outcomes by reducing weight loss and mortality, while modulating T cell responses through increased Tregs and decreased IFN-γ/granzyme B-producing T cells. These findings support the clinical exploration of anti-IL-21 antibodies for GVHD prophylaxis [68]. While IL-13 demonstrates protective effects in GVHD animal models, clinical observations reveal a paradoxical association: elevated donor-derived IL-13 levels prior to transplantation strongly predict severe acute GVHD development in patients, suggesting type 2 cytokines (IL-5/IL-13) may critically influence graft outcomes during cytokine storms [181]. Furthermore, IL-13 exhibits chemotactic effects on monocytes and eosinophils. The data indicate that IL-13 from donor T cells during aGVHD may directly contribute to the pathology of this inflammatory response [178]. In GVHD animal models, IL-26 promotes inflammation by enhancing Th17 responses and tissue fibrosis. A newly developed humanized anti-IL-26 monoclonal antibody effectively reduced T cell/neutrophil infiltration and fibroproliferation, demonstrating therapeutic potential for chronic GVHD treatment [40]. Meanwhile, several clinical studies have reported that patients with GVHD and IBD exhibit significantly elevated IL-26 mRNA expression in their blood or lesions. Furthermore, when neutralizing IL-22 in cutaneous GVHD models, reductions were observed in inflammation severity, skin epidermal thickness, and fibrosis. This suggests that during chronic inflammation conditions like GVHD, IL-22 may play a role that promotes disease progression rather than amelioration, fezakinumab treatment has been shown to improve clinical disease scores in patients with atopic dermatitis, particularly among those with elevated serum IL-22 levels [182]. Currently, IL-22-IgG2 fusion proteins are under evaluation for the treatment of acute lower gastrointestinal GVHD [180]. IL-11 has been shown to reduce cytokine release and enhance survival rates in mouse models of bone marrow transplantation. IL-11 can mitigate mucositis by decreasing intestinal permeability, partially shifting T cells towards the Th2 phenotype, down-regulating IL-12 levels, and promoting the recovery of oral and intestinal mucosa [100,183]. Despite demonstrating promising aGVHD effects in animal studies, its efficacy and safety in clinical trials remain contentious. Consequently, IL-11 is not recommended as a preventive measure for GVHD in allogeneic transplantation settings; further investigation into its clinical applications is warranted [184]. Low-dose IL-2 therapy has been shown to expand Tregs, thereby alleviating chronic GVHD. Concurrently, clinical trials have demonstrated the safety of ustekinumab in HCT recipients [87]. These findings position IL-2-mediated Treg enhancement and IL-12/23 pathway inhibition as complementary strategies for GVHD control, warranting further investigation of combination therapies with IL-12/15/18-activated NK cells [185].

## 6. Prospects and Future Directions

Research on the interleukin family in the context of GVHD is poised to deepen, unveiling further complexities regarding disease pathogenesis and therapeutic strategies. As advancements in biotechnology and immunological research continue, we anticipate a more precise understanding of the specific mechanisms by which interleukin family members contribute to GVHD, as well as their intricate interactions with the host immune system, gut microbiota, and other environmental factors. In terms of basic research, the application of single-cell multi-omics technologies will enable a detailed characterization of IL-secreting cells within GVHD target organs (e.g., gut and liver) [186]. Furthermore, by integrating epigenetics and metabolomics approaches, future studies are expected to focus increasingly on uncovering regulatory networks among interleukin family members and their dynamics across different phases of GVHD (e.g., acute versus chronic phases) [187]. This comprehensive understanding will provide a theoretical foundation for developing more targeted therapeutic interventions. Regarding clinical applications, targeted therapies aimed at interleukin family members represent an important avenue for treating GVHD [188]. For example, developing monoclonal antibodies or small molecule inhibitors against specific interleukins (e.g., IL-1β, IL-6, IL-17) could effectively inhibit pro-inflammatory cytokine activity and subsequently mitigate the severity of GVHD. Concurrently, enhancing the function of anti-inflammatory cytokines (e.g., IL-10 and IL-22) may also emerge as a novel strategy for managing this condition. In addition, advancements in immune cell therapy and gene editing technologies are poised to enhance the personalization of GVHD treatment. For instance, employing CRISPR-Cas9 to modify donor T cells can diminish their pro-inflammatory potential, while enhancing immune regulation through the expansion of Tregs in vitro combined with IL-2 treatment. With the rapid evolution of these therapeutic modalities, future approaches to GVHD management will be increasingly personalized and precise. For example, gene editing technology may enable modifications of a patient’s immune cells to more effectively inhibit the onset and progression of GVHD; alternatively, augmenting the patient’s anti-tumor capabilities and immune tolerance could be achieved by over-transferring genetically modified or cytokine-pre-treated immune cells. We anticipate that a more comprehensive system comprising animal models and clinical trials for GVHD will be established in the future, facilitating a more accurate assessment of both efficacy and safety for novel therapies. This development is expected to provide a robust experimental foundation and clinical evidence supporting the application of interleukin family members in GVHD treatment, ultimately leading to more effective and safer therapeutic options for patients suffering from this condition.

In summary, the research of interleukin family in the field of GVHD is full of challenges and opportunities. With the deepening of research and the continuous progress of technology, we have reason to believe that in the future we will be able to develop more accurate and effective treatments for GVHD, bringing better prognosis and quality of life to patients.

## Figures and Tables

**Figure 1 ijms-26-08620-f001:**
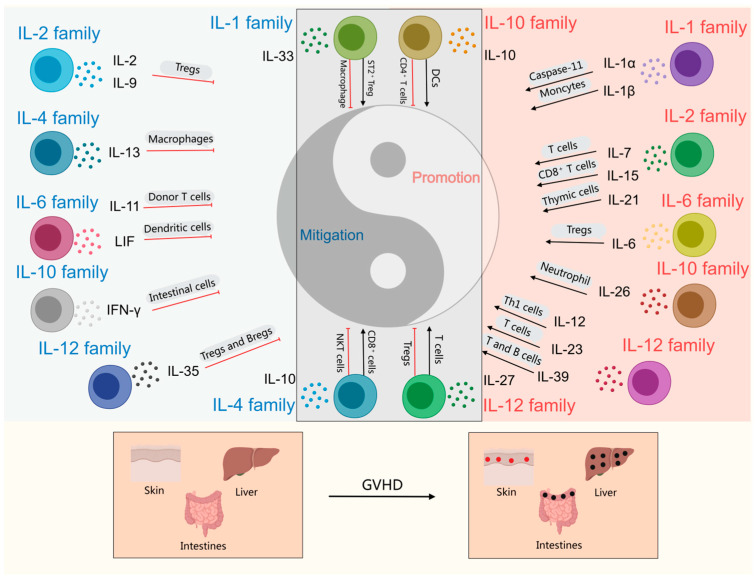
The interleukin network in GVHD. (Created with MedPeer, medpeer.cn). On the left side of the figure is “mitigation”, which is indicated by the red arrow. On the right is “promotion”, represented by a black arrow indicating promotion. The middle part includes both “mitigation” and “promotion”, represented by a red and black arrow, respectively, to indicate inhibition or promotion. Specific interleukins within the interleukin family influence the development and progression of GVHD by modulating the functions of their target immune cells.

**Table 1 ijms-26-08620-t001:** Interleukins promoting the progression of GVHD.

	Interleukin	Disease Promotion/Mitigation	Mode of Action	References
IL-1 family	IL-1α IL-1β	Promotion	1. Caspase-11 signaling enhances GVHD via IL-1α. 2. IL-1β is produced by monocytes cells, affects DCs and T cells, and plays a key pro-inflammatory role in the early stages of GVHD, especially Th17 cells.	[27,28,29,30]
IL-2 family	IL-7	Promotion	IL-7 also promotes the expansion of allogeneic reactive T cells that mediate GVHD.	[31,32]
	IL-15	Promotion	1. IL-15 significantly increased tissue inflammation in the gut and liver as well as GVHD morbidity and mortality after BMT by promoting the expansion and activation of allogeneic reactive effector memory CD8+ T cells.	[33,34]
	IL-21	Promotion	IL-21 directly stimulates the development of T cells in the thymus by increasing the number of thymic progenitor cells and promoting the recovery of TEC, IL-21 promotes Th17 differentiation in the presence of TGF and induces inflammation.	[35]
IL-6 family	IL-6	Promotion	1. IL-6 facilitates the differentiation of Th17 cells in conjunction with TGF-β, while concurrently inhibiting the differentiation of Treg cells induced by TGF-β. 2. Abnormal expression of IL-6 drives the occurrence and progression of cGVHD by promoting macrophage differentiation and maturation, TGF-β production and tissue fibrosis.	[36,37,38].
IL-10 family	IL-26	Promotion	1. IL-26 activates mouse fibroblasts, promotes collagen production, and aggravates GVHD pulmonary fibrosis. 2. IL-26 significantly increases neutrophil levels in GVHD target tissues and peripheral blood. The systemic symptoms of GVHD were exacerbated by significantly elevated levels of Th17 cytokine expression in donor CD4+ T cells, and significantly enhanced levels of granulocyte colony-stimulating factor, IL-1β and IL-6.	[39,40]
IL-12 family	IL-12	Promotion	IL-12 activates the STAT3 and STAT4 signaling pathways, facilitating the differentiation of naïve T cells into Th1 cells, while significantly enhancing the production of IFN-γ. Concurrently, it inhibits the synthesis of Th2 cytokines such as IL-4, thereby establishing a Th1-dominated immune response.	[41]
	IL-23	Promotion	1. IL-23 plays a key role in the pathological damage of colonic GVHD mainly through T cell-dependent pathways, especially by inducing IFN-γ secretion. 2. IL-23 may affect aGVHD through IL-17-dependent and non-dependent pathways.	[42,43]
	IL-39	Promotion	IL-39 promotes pro-inflammatory responses in T and B cells and plays an important role in the pathophysiology of cGVHD by activating the STAT pathway through interaction with IL-39 receptors on T cells.	[44]

**Table 2 ijms-26-08620-t002:** Interleukins inhibit the progression of GVHD.

	Interleukin	Disease Promotion/Mitigation	Mode of Action	References
IL-2 family	IL-2	Mitigation	1. IL-2, by pairing Tregs with oIL-2, is able to selectively expand Tregs in vivo while avoiding the activation of allogeneic reactive T cells, thereby reducing the severity of aGVHD and improving survival. 2. Increase the expression of PD-L1 in host tissues by blocking the binding of IL-2 to the IL-2 receptor on T cells using the anti-IL-2 monoclonal antibody JES6, and inhibit pathogenic T cell function by inhibiting the activation of the IL-2-Stat5 signaling pathway in donor T cells to promote T cell depletion or differentiation into Tr1 cells.	[92,93]
	IL-9	Mitigation	1. IL-9 activates Treg cell recruitment and activates mast cell-mediated regional immunosuppression. 2. Th9 cells secreting IL-9 inhibit type I cytokine production by allogeneic-reactive T cells while maintaining IL-9 secretion capacity and suppressing IFN-γ-driven allogeneic responses.	[19,94]
IL-4 family	IL-13	Mitigation	1. IL-13 Cultures produce MDSC-IL-13, which plays a protective role in GVHD by up-regulating the expression of arginase-1, which is more inhibitory to allogeneic T cell responses. 2. IL-13 can inhibit the expression of pro-inflammatory cytokines such as IL-1β, TNF-α, and IL-6 in monocytes and macrophages, which contributes to reducing tissue damage. 3. IL-13 promotes the shift in Th1/Th2 balance to Th2 cells by inhibiting TNF-α and enhancing the secretion of IL-4 and IL-5, thus having a protective function in GVHD.	[95,96,97,98]
IL-6 family	IL-11	Mitigation	IL-11 affects the activation of donor T cells by down-regulating IL-12, which directly inhibits the production of inflammatory cytokines such as TNF-α and IFN-γ by monocytes and macrophages, providing a new strategy for the treatment of GHVD.	[99,100]
	LIF	Mitigation	rLIF activates the STAT1 signaling pathway, resulting in the downregulation of IL-12-p40 expression in irradiated recipient dendritic cells. This process leads to a decrease in MHC class II expression on intestinal epithelial cells, which subsequently reduces donor T cell activation and infiltration.	[101,102]
IL-10 family	IFN-λ	Mitigation	IFN- λ limits the loss of intestinal stem cells and promotes the regeneration of intestinal epithelial cells, thereby protecting the mucosal barrier function. IFN- λ is a key protective factor in the immunopathology of GVHD in the gastrointestinal tract and plays an important role especially in the ISC compartment.	[103]
IL-12 family	IL-35	Mitigation	IL-35 is mainly released by Treg and Breg and plays a role in suppressing inflammation and reducing the severity of autoimmune diseases.	[104]
IL-17 family	IL-25	Mitigation	IL-25 protects cuprocytes from GVHD, prevents bacterial translocation, reduces IFN-γ and IL-6 plasma levels, and ameliorates GVHD.	[105]

**Table 3 ijms-26-08620-t003:** Interleukins with dual effects on GVHD progression.

	Interleukin	Disease Promotion/Mitigation	Mode of Action	References
IL-1 family	IL-33	Promotion/mitigation	1. The IL-33/ST2 axis exacerbates the severity of GVHD by increasing IFN-g production, upregulating IL-18R expression and promoting cell proliferation. 2. IL-33’s can promote tissue repair by amplifying Tregs, especially the ST2^+^ Treg subpopulation, and by up-regulating the expression of bi-regulated proteins. 3. IL-33 functions by inhibiting M1-type macrophage activation and promoting granulocyte MDSC.	[133,134,135,136]
	IL-18	Promotion/mitigation	1. In mouse model studies, IL-18 pre-activated NK cells were able to maintain Eomes and T-bet expression and inhibit acute GVHD. 2. IL-18 administration resulted in an increase in the number of CD8^+^ T cells and a significant decrease in the number of Treg cells, an effector that accelerated the onset and exacerbated the severity of GVHD. 3. IL-18 exerts its therapeutic effect on chronic GVHD through a triple mechanism of action by inducing donor-versus-host CD8^+^ CTLs, down-regulating host B cell MHC class II expression, and reducing the number of donor CD4^+^ T cells, etc.	[85,137,138]
IL-4 family	IL-4	Promotion/mitigation	1. IL-4 derived from basophils, NKT cells, and conventional CD4^+^ T cells can effectively inhibit GVHD, and exogenous IL-4 treatment also demonstrates alleviating effects on acute GVHD. 2. The proportion of IL-4-producing CD8^+^ T cells in the peripheral blood of cGVHD patients was significantly higher than in non-cGVHD patients and healthy controls, suggesting these cells may serve as an immunological marker for cGVHD.	[95,112,139]
IL-10 family	IL-10	Promotion/mitigation	1. ROCK1/2 inhibitors reduce CD8^+^ cell proliferation by inhibiting DCs and increase the expression of IL-10 mRNA, which plays an anti-inflammatory role during aGVHD. 2. When co-administered with IL-2, IL-10 initially inhibits activation and expansion of responding T cells, but later modulates the response to induce massive oligoclonal expansion of CD4^+^ T cells, ultimately leading to GVHD lethality.	[37,140]
	IL-22	Promotion/mitigation	1. IL-22 expressed in ILC3 cells from the intestine and thymus supports intestinal epithelial and thymic regeneration and barrier function during allografting, respectively. 2. IEC can secrete Reg3α and Reg3γ in response to IL-22 stimulation, and elevated serum Reg3α levels may reflect more severe intestinal barrier damage, predisposing patients to more severe intestinal GVHD. 3. The synergistic effect between IFN-α and IL-22 activates STAT1 in colonocytes, while prompting CXCL10 to recruit CXCR3-expressing effector T cells to the site of tissue damage, exacerbating GHVD.	[141,142,143]
IL-12 family	IL-27	Promotion/mitigation	1. Inhibition of IL-27 selectively reduced IL-10 production in conventional T cells without affecting IL-10 production in Treg cells, thus preserving the ability of Treg to inhibit GVHD through this mechanistic pathway. 2. IL-27 in the inflammatory milieu in turn regulates the function of hPMSCs, upregulates PDL2 expression in hPMSCs via the JAK/STAT pathway, which in turn enhances the ability to induce CD4^+^ IL-10^+^ IFN-g^+^ T cell production and has therapeutic implications for the control of GVHD.	[144,145]
IL-17 family	IL-17	Promotion/mitigation	1. IL-17 can protect against GVHD by inhibiting IL-12 production by donor macrophages to downregulate the Th1 response. 2. IL-17A is a key part of the pathogenicity of these Th17 cells in GVHD.	[130]

**Table 4 ijms-26-08620-t004:** Role of interleukins in clinical trials of GVHD.

Interleukin	Intervention Methods	Mechanisms of Action	References
IL-7/IL-5	Prognostic biomarkers for acute GVHD	Within the first month after BMT, elevated IL-7 and IL-15 levels independently predicted the occurrence and severity of aGVHD.	[177]
IL-21	Monoclonal antibody	The T cell response was regulated by increasing the number of regulatory T cells (Tregs) and reducing the number of T cells that produce interferon γ/granzyme B.	[68]
IL-13	Prognostic molecules for aGVHD	IL-13 has a chemotactic effect on monocytes and eosinophils.	[178]
IL-26	Monoclonal antibody	Reducing T cell/neutrophil infiltration and fibroblast proliferation shows potential for the treatment of chronic GVHD.	[40]
IL-22	1.Monoclonal antibody(fezakinumab) 2.Recombinant fusion protein(F-652)	1.After neutralization of IL-22, the severity of inflammation, epidermal thickness and fibrosis were reduced 2.F-652 plays an important protective role in promoting tissue survival and regeneration under immune attack.	[179,180]
IL-11	Recombinant protein	IL-11 alleviates inflammation by reducing the transformation of T cells to Th2 phenotype, downregulating IL-12 levels and promoting mucosal repair.	[100]
IL-2	Recombinant protein	Low-dose IL-2 treatment has been shown to amplify Tregs and thus alleviate cGVHD.	[87]

## Data Availability

No data was used for the research described in the article.

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
