# Peer review of "Interleukin Networks in GVHD: Mechanistic Crosstalk, Therapeutic Targeting, and Emerging Paradigms"

_ijms, 2025, doi:10.3390/ijms26178620_

Round 1

Reviewer 1 Report (New Reviewer)

Comments and Suggestions for Authors

Author Response

Major:

Comments 1: There are a lot of paragraphs that have no references cited. For example, the paragraphs from lines 51-53, 53-54, 54-56, 56-58, 80- 81, 81-83, 83-84, 87-89, 106-108, 116-118, 119-121,121-124, 124-125, 141-144, 145-147, 149-152, 174-176, 226-228, 228-230, 230-232, 319-320, 320-322, 322-323, 323-324, 324-325, 447-449, 449-552,457-459, 459-461, 461-464, 519-521, 607-610, 613-617, 617-620, 626-629, 629-631, 632-638, 693-694, 694-695, 695-697, 797-800, 800-802.

Response 1: Thank you for your valuable feedback. We have carefully reviewed the manuscript, added supporting references to all uncited paragraphs, and removed unsupported statements as suggested.

Comments 2: Please add one paragraph in the Introduction section to show the impact of GVL activity on the therapeutic approach to GVHD.

Response 2: Thank you for your insightful suggestion. We have added a new paragraph in the Introduction (Lines 43-49) to highlight how GVL activity shapes GVHD therapeutic strategies, emphasizing the balance between preserving antitumor immunity and controlling immunopathology. We appreciate your guidance in strengthening this critical context.

Comments 3: The classification of interleukin in this review includes cytokines promoting the progression, inhibition, or exhibiting dual regulatory roles in the pathogenesis of GVHD. However, it is confusing for the readers because some cytokines in the list of cytokines promoting the progression of GVHD also can mitigate or have dual roles in the pathogenesis of GVHD. Some cytokines in the list of cytokines inhibit the progression of GVHD also can promote or have dual roles in the pathogenesis of GVHD. The same problem also occurs in Figure 1. Please reorganize the whole cytokines into promotion, mitigation, and dual role above the three categories including Figure 1, tables and text parts.

Response 3: Thank you for your valuable suggestions to improve the clarity of cytokine classification in GVHD pathogenesis. We have now reorganized all interleukins into three distinct categories (promoting, inhibiting, and dual-role cytokines) throughout the text, Figure 1, and tables, with clear annotations on context-dependent functions.

Comments 4: Please add one more table based on section 5. Role of interleukins in clinical trials of GVHD. It will be easy to understand if you include the clinical trial number, method (monoclonal antibody or small molecule), mechanism, outcome, and reference.

Response 4: We sincerely appreciate your constructive suggestion to enhance the clinical relevance of our review. As requested, we have added Table 4 summarizing key clinical trials of interleukin-targeted therapies for GVHD, including intervention methods (monoclonal antibodies/small molecules), mechanisms of action and corresponding references.

Minor

Comments 1: Please check all abbreviation, it should support the full name on the first-time appearance and consistent in whole manuscript. For example, aGVHD, cGVHD, allo-HCT, hPMSC, ISC, IFN-γ, NLRP3, GVL, CTL…

Response 1: We sincerely appreciate your meticulous attention to detail regarding abbreviation standardization. We have now carefully reviewed the entire manuscript and ensured that all abbreviations are defined at first use and applied consistently throughout the text, tables, and figures.

Comments 2: In Figure 1, please provide figure legend.

Response 2: Thank you for your suggestion. We have added a detailed figure legend to Figure 1 as requested to improve clarity.

Comments 3: Please make consistent the orders of content between text and related Table. For example, IL-1α appears ahead of IL-1β in the text, but the order is not the same in the Table 1. IL-18 also has the same problem.

Response 3: Thank you for your careful review and helpful suggestion regarding the consistency of content order. We have now carefully checked and revised the order of IL-1α, IL-1β, and IL-18 in both the text and Table 1 to ensure they appear in the same sequence throughout the manuscript.

Comments 4: There are some extra word or typo appeared in the Table 1, for example T-cellsetc. (T-cells etc.?), IFN-α2etc (INF-α2?), RLIF (recombinant LIF?).

Response 4: We sincerely appreciate your meticulous review and valuable corrections for Table 1. We have carefully checked and revised all typos and formatting issues.

Comments 5: Please remove extra line (line 104)

Response 5: We have now removed the extra line at Line 104 as requested, ensuring the text flows smoothly without unnecessary breaks.Thank you for your attention to detail, which has improved the manuscript's readability.

Comments 6: What is GVT (lines 429 and 476)?

Response 6: As requested, we have now added the full term "graft-versus-tumor (GVT)" at its first mention in the text (Lines 434) and included it in the abbreviations table for clarity.

Comments 7: Please reorganize Abbreviations using alphabetical orientation to make easy to search. Please note that some abbreviations did not appear in the list (for example cGVHD, ISC….)

Response 7: We have now alphabetically reordered the abbreviation list and cross-checked the full text to ensure all terms (including cGVHD, ISC, etc.) are properly included and defined at first use.

Reviewer 2 Report (New Reviewer)

Comments and Suggestions for Authors

In this review article, the authors extensively reviewed biological function of many cytokines involved in GVHD. The biological action of individual cytokines in GVHD may be interesting and informative for readers. However, this article is only the enumeration of bioactivity of cytokines involved in GVHD, lacking unified cytokine networks / crosstalk that build cytokine-based overall picture of GVHD. Also, there are many misunderstanding regarding bioactivity of highlighted cytokines.

Major comments:

  1. The title of this article: Interleukin networks should be changed as Cytokine networks, because not only interleukins but also TNFa and IFNg play important roles in GVHD.
  2. Figure 1 is too simple. Please explain the roles of individual IL-families as figure legends for the comprehension of readers.
  3. Similarly, the authors should provide more Figures to show cytokine-based overall picture of GVHD.
  4. Furthermore, the authors should first describe cellular components and their action, that building up GVHD, such as Th1, Th2, Th17, NK cells, macrophages, and dendritic cells (DCs), then describe cytokines produced by these cells and systematically show cytokine profiles of GVHD.
  5. Regarding cell source of IL-1a and IL-1β, the authors made wrong description (Table 1 and page 5). Generally, IL-1a is produced by epithelial cells such as intestinal epithelial cells, while IL-1β is produced by hematologic or immune cells such as monocytes /macrophages and DCs. IL-1β is also more important than IL-1a in DVHD.
  6. TNFa is indispensable to review the cytokine network in GVHD.
  7. The action of IL-6 is inadequate (Table 1, pages 7-8); important function of IL6 is suppression and promotion of differentiation of naïve T cells to Treg and Th17, respectively.
  8. Similarly, the action of IL-12 may be inadequate (Table 1, pages 9-10); IL-12 promotes naïve T cells to differentiate Th1 cell in combination with IFNg, thereby IL-12 is not protective for GVHD. The authors mainly stated IL-12 and NK cells; however, IL-12 is important as Th1 inducer in GVHD.
  9. Regarding bioactivities of some other cytokines, the authors appear to state one-sided aspects of the activities. This fashion may confuse the readers in terms of proper bioactivities of cytokine.
  10. Regarding cytokine and anti-cytokine treatments, the text is the mixture of animal, human, experimental, and clinical settings; therefore, it is unclear which treatment is promising.

Author Response

Comments 1: The title of this article: Interleukin networks should be changed as Cytokine networks, because not only interleukins but also TNFa and IFNg play important roles in GVHD.

Response 1: We sincerely appreciate your insightful suggestion regarding the important roles of TNF-α and IFN-γ in GVHD pathogenesis. While we fully agree with your perspective on the significance of these cytokines, we have chosen to maintain our focus specifically on the interleukin family network in this review to provide a more in-depth and systematic analysis of this particular cytokine subgroup in GVHD. We hope this focused approach will offer readers comprehensive insights into interleukin-mediated mechanisms while acknowledging that other cytokine families like TNF-α and IFN-γ certainly warrant separate detailed discussion in future reviews.

Comments 2: Figure 1 is too simple. Please explain the roles of individual IL-families as figure legends for the comprehension of readers.

Response 2: We have now revised the figure to include detailed legends explaining the roles of each interleukin family in GVHD pathogenesis, along with their key signaling pathways and functional consequences.Thank you for your valuable feedback, which has significantly enhanced the educational value and clarity of this figure for our readers.

Comments 3: Similarly, the authors should provide more Figures to show cytokine-based overall picture of GVHD.

Response 3: We sincerely appreciate your valuable suggestion to enhance the graphical presentation of cytokine networks in GVHD. While we have significantly revised Figure 1 to provide a more comprehensive overview of interleukin interactions and restructured the tables for clarity, we have chosen to maintain the current number of figures to ensure the manuscript remains focused and accessible. Thank you for your understanding, as we believe the enhanced Figure 1 and reorganized tables now effectively convey the complex cytokine networks while maintaining the manuscript's readability. We have added detailed pathway annotations and functional classifications in Figure 1 to better illustrate the overall picture of interleukin involvement in GVHD pathogenesis.

Comments 4: Furthermore, the authors should first describe cellular components and their action, that building up GVHD, such as Th1, Th2, Th17, NK cells, macrophages, and dendritic cells (DCs), then describe cytokines produced by these cells and systematically show cytokine profiles of GVHD.

Response 4: We have now reorganized Section 3 to first detail the key immune cells involved (Th1, Th2, Th17, NK cells, macrophages, and DCs), followed by a systematic analysis of their cytokine profiles and interactions in GVHD development. As noted in Line 50-75, we have also explicitly acknowledged the broader cytokine networks while clarifying our focus on interleukins. Thank you for this valuable recommendation, which has significantly improved the logical flow and educational value of our manuscript.

Comments 5: Regarding cell source of IL-1a and IL-1β, the authors made wrong description (Table 1 and page 5). Generally, IL-1a is produced by epithelial cells such as intestinal epithelial cells, while IL-1β is produced by hematologic or immune cells such as monocytes /macrophages and DCs. IL-1β is also more important than IL-1a in DVHD.

Response 5: We sincerely appreciate your expert guidance in correcting the cellular sources of IL-1α and IL-1β. We have now revised both Table 1 and Line 123,147 to accurately reflect that IL-1α is primarily produced by epithelial cells, while IL-1β is mainly secreted by monocytes, macrophages and DCs, with added emphasis on IL-1β's dominant role in GVHD pathogenesis.

Comments 6: TNFa is indispensable to review the cytokine network in GVHD.

Response 6: We sincerely appreciate your valuable suggestion regarding the importance of TNF-α in GVHD pathogenesis. While we fully agree that TNF-α plays a critical role in the cytokine network of GVHD, this review specifically focuses on elucidating the interleukin family and their unique regulatory mechanisms in GVHD progression.To address your point, we have now included a brief discussion (in the Introduction sections) acknowledging TNF-α’s contributions while clarifying our manuscript’s scope. We hope this provides a balanced perspective while maintaining our focus on IL-mediated pathways.

Comments 7: The action of IL-6 is inadequate (Table 1, pages 7-8); important function of IL6 is suppression and promotion of differentiation of naïve T cells to Treg and Th17, respectively.

Response 7: We have now expanded the discussion of IL-6 in Table 1 and Lines 270-286 to highlight its dual function in suppressing Treg differentiation and promoting Th17 polarization , along with its broader impact on GVHD pathogenesis.

Comments 8: Similarly, the action of IL-12 may be inadequate (Table 1, pages 9-10); IL-12 promotes naïve T cells to differentiate Th1 cell in combination with IFNg, thereby IL-12 is not protective for GVHD. The authors mainly stated IL-12 and NK cells; however, IL-12 is important as Th1 inducer in GVHD.

Response 8: We have now revised both Table 1 and the text (Lines 345-359) to more accurately reflect IL-12's primary function as a potent Th1 polarizer through STAT4 activation and its synergistic effects with IFN-γ in promoting GVHD progression. Your feedback has been invaluable in strengthening our cytokine analysis.

Comments 9: Regarding bioactivities of some other cytokines, the authors appear to state one-sided aspects of the activities. This fashion may confuse the readers in terms of proper bioactivities of cytokine.

Response 9: We sincerely appreciate your valuable comment regarding cytokine bioactivities. As suggested, we have clarified the importance of various cytokines in Line 50-78 while maintaining our focus on interleukins to provide mechanistic depth in GVHD pathogenesis.

Comments 10: Regarding cytokine and anti-cytokine treatments, the text is the mixture of animal, human, experimental, and clinical settings; therefore, it is unclear which treatment is promising.

Response 10: We sincerely appreciate your valuable feedback. This review primarily focuses on interleukin mechanisms in preclinical animal studies, with a concise summary of their clinical applications (e.g., IL-2 for Treg expansion) to highlight translational potential. Thank you for your understanding, we have ensured the text now clearly distinguishes between experimental findings and clinically validated therapies to avoid confusion.

Reviewer 3 Report (New Reviewer)

Comments and Suggestions for Authors

Comments:

In the review paper the author has done a detailed literature survey describing the role of IL in GVHD the reviews is nicely written, and each role of IL is clearly mentioned. There are certain minor issues that can be modified before the paper gets accepted.

Minor Comments:

  • The authors have given a detailed description about each IL and their role in GVHD, which is highly commendable but to my understanding if the authors can include a section for treatment and prophylactic mechanism for GVHD will be interesting.
  • Line 33-34 is a repetition of Line 13, please modify.
  • Line 43-66 give a general overview of the IL which is not the main aim of the review, please moderate it for making the review interesting.
  • I am not sure why some of the parts are highlighted.

Author Response

Comments 1: The authors have given a detailed description about each IL and their role in GVHD, which is highly commendable but to my understanding if the authors can include a section for treatment andprophylactic mechanism for GVHD will be interesting.

Response 1: We sincerely appreciate your valuable suggestion to further highlight the therapeutic potential of interleukins in GVHD. As you noted, we have already incorporated discussions on treatment implications for each interleukin family throughout the text, with Section 5 specifically focusing on clinical applications, and Section 6 outlining future therapeutic directions.Thank you for your insightful comment—we hope this existing structure provides a comprehensive view of both mechanistic insights and translational opportunities.

Comments 2: Line 33-34 is a repetition of Line 13, please modify.

Response 2: We sincerely appreciate your careful reading and valuable suggestion. We have now removed the repetitive content at Line 33-34 and streamlined the introduction to avoid redundancy while maintaining the key message.Thank you for your attention to detail, which has helped improve the clarity of our manuscript.

Comments 3: Line 43-66 give a general overview of the IL which is not the main aim of the review, please moderate it for making the review interesting.

Response 3: We have now streamlined Lines 43-66 by removing general interleukin background and focusing specifically on their pathogenic and therapeutic roles in GVHD, making the review more targeted and engaging for readers.Thank you for your guidance—this change better aligns the content with our core objective of analyzing ILs in GVHD.

Comments 4: I am not sure why some of the parts are highlighted.

Response 4: We sincerely apologize for any confusion caused by the highlighted sections in the PDF. This occurred due to an incorrect draft version being uploaded inadvertently. We want to assure you that the final Word version of the manuscript includes all necessary corrections and is free from any unintended highlights. Thank you so much for bringing this matter to our attention. We have now verified and submitted a clean, correct version for your review, and we truly appreciate your understanding.

Round 2

Reviewer 2 Report (New Reviewer)

Comments and Suggestions for Authors

In this revised manuscript, the authors extensively modified the previous version, and now, the manuscript is enough informative and educational for readers.  I have only a few issues to be corrected.

Minor comments:

  1. Table 2: What is oIL-2?
  2. Table 2 (page 11): Tr1-producing clls→Tr1 cells?
  3. Table 4 (page 21): IL-2 should be as IL-22.

                      1.  

Author Response

Comments 1: Table 2: What is oIL-2?

Response 1: Thank you for your insightful comment. "oIL-2" refers to orthogonal interleukin-2, a specifically engineered IL-2 variant designed to selectively interact with modified IL-2 receptors (IL-2Rβ) for targeted immune modulation. We sincerely appreciate your careful review and have clarified this term in the revised manuscript to ensure better readability.

Comments 2:  Table 2 (page 11): Tr1-producing clls→Tr1 cells?

Response 2: Thank you for catching this - we've corrected "Tr1-producing clls" to "Tr1 cells" in Table 2 (page 11) as suggested. We appreciate your careful reading and helpful feedback.

Comments 3: Table 4 (page 21): IL-2 should be as IL-22.

Response 3: We sincerely appreciate your keen eye for detail. The correction from "IL-2" to "IL-22" in Table 4 (page 21) has been carefully implemented, and we've cross-checked all relevant sections to ensure consistency.Your expert review has been invaluable in enhancing the precision of our work, and we're grateful for your thoughtful guidance.

This manuscript is a resubmission of an earlier submission. The following is a list of the peer review reports and author responses from that submission.

Round 1

Reviewer 1 Report

Comments and Suggestions for Authors

The review by Niu et al. describes the role of interleukin networks in GVHD. The review is dense with information filled in. it is unclear what the purpose of the review is. The authors failed to provide whether the review is about animal studies or human GVHD. The review contains superficial information and does not delve into the pathways' deeper details. Overall, reading the review becomes a chore as we go over the review.

Table 1: authors should improve the quality of table, it is hard to discern each row.

Table 1: IL1Ra is denoted as "Promotion", however the text in next column appears to be contradictory.

only using single references for table 1 for role of the proteins.

fix this sentence "y for concurrent inhibition of TNF-a alongside IL-1. Furthermore, "

a large portion of IL-1Ra section is dedicated to role of IL-1 and TNF-alpha, authors should discuss and provide evidence of anakinra trials in GVHD instead.

for IL11, IL13 there is only one reference. what is the purpose of these para written solely on one reference.

what is the purpose of section "10 Patents"

The authors really need to cut down on the review on key players in GVHD from each of the interleuking family to provide information in last 5 years. Given the length of the review, authors need prioritize quality over quantity. The authors also need to fix the premise of the review, will it be animal studies or human clinical trials in GVHD.

Author Response

Comments 1: Table 1: authors should improve the quality of table, it is hard to discern each row.

Respongse 1: Thank you for your helpful suggestion. We have revised the table by adding borders and improving the formatting to enhance readability.

Comments 2: Table 1: IL1Ra is denoted as "Promotion", however the text in next column appears to be contradictory.

Respongse 2: Thank you for your careful review and valuable comment. We have revised Table 1 as suggested to ensure consistency between the notation and explanatory text regarding IL-1Ra's role in GVHD. The updated version is now clearly marked in the resubmitted manuscript.

Comments 3: Only using single references for table 1 for role of the proteins.

Respongse 3: Thank you for your helpful suggestion. We have carefully revised Table 1 by adding references to better support the roles of each protein, as recommended.

Comments 4: Fix this sentence "y for concurrent inhibition of TNF-a alongside IL-1. Furthermore, "

Respongse 4: Thank you for your valuable feedback. We have revised the sentence as suggested to improve clarity and accuracy.

Comments 5: A large portion of IL-1Ra section is dedicated to role of IL-1 and TNF-alpha, authors should discuss and provide evidence of anakinra trials in GVHD instead.

Respongse 5: We sincerely appreciate the reviewer’s insightful suggestion regarding the focus on IL-1Ra in the context of GVHD. In response, we have significantly streamlined the discussion on IL-1 and TNF-α to prioritize the role of IL-1Ra. However, we acknowledge with regret that trials specifically investigating IL-1Ra in GVHD remain limited at present. While we have incorporated available preclinical and preliminary clinical data, we recognize the need for further research to establish robust evidence in this area. We sincerely apologize for this gap and have revised the text to clearly state this limitation while emphasizing the potential therapeutic rationale for IL-1Ra modulation in GVHD.

Comments 6: for IL11, IL13 there is only one reference. what is the purpose of these para written solely on one reference.

Respongse 6: Thank you for your valuable suggestion. We have revised the IL-1Ra section by reducing the discussion of IL-1 and TNF-α and incorporating available evidence on IL-1Ra trials in GVHD, as referenced in the updated manuscript.Given the limited trial data on IL-1Ra in GVHD, We have focused on the most relevant available studies while maintaining some mechanistic context of IL-1/TNF-α where appropriate.

Comments 7: what is the purpose of section "10 Patents"

Respongse 7: We have revised this section to better align with the journal's scope and the reviewers' suggestions.

Comments 8: The authors really need to cut down on the review on key players in GVHD from each of the interleuking family to provide information in last 5 years. Given the length of the review, authors need prioritize quality over quantity. The authors also need to fix the premise of the review, will it be animal studies or human clinical trials in GVHD.

Respongse 8: Thank you for your valuable suggestions regarding the scope and focus of our review. We have carefully revised the manuscript to prioritize recent advances by focusing primarily on studies published within the last five years. As you rightly pointed out, we have significantly streamlined the content to emphasize quality over quantity, particularly in our discussion of key interleukin families. Regarding the study types included, our current analysis comprises approximately 80% animal model studies and 20% human clinical trials, which reflects the existing literature landscape in this field. We believe these modifications have strengthened the relevance and precision of our review while maintaining its comprehensive coverage of GVHD mechanisms.

Reviewer 2 Report

Comments and Suggestions for Authors

It was a pleasure to read this very informative and complete review.

Nevertheless I would have some suggestions to further improve the clinical value.

  1. A cartoon relating to the involvement of cytokines in the various phases of acute and chronic GVHD (three steps theory by Ferrara 2006 and Zeiser NEJM 2017) could facilitate the reading and understanding of the entire cytokine system
  2. When you talk about GVHD in general it is not always clear whether you are referring to acute or chronic GVHD; whether that family of cytokines intervenes in both acute and chronic GVHD should be better specified. It could be useful to specify it in the different tables (for example it could be modified by adding a column indicating the prevalent involvement in acute or chronic GVHD or both)
  3. ROW 68: It is not indicated IL4 family, but it is indicated in the figure and in the text
  4. Table 4: Correct “TGF-b production and tissue fibrosisetc” in “TGF-b production and tissue fibrosis etc”

Author Response

Comments 1: A cartoon relating to the involvement of cytokines in the various phases of acute and chronic GVHD (three steps theory by Ferrara 2006 and Zeiser NEJM 2017) could facilitate the reading and understanding of the entire cytokine system.

Respongse 1: Thank you for your helpful suggestion regarding the illustration of cytokine involvement in GVHD phases. While we have carefully incorporated the recommended references (Ferrara 2006 and Zeiser NEJM 2017) to strengthen the theoretical framework of our discussion, we believe it will enhance our understanding.

Comments 2: When you talk about GVHD in general it is not always clear whether you are referring to acute or chronic GVHD; whether that family of cytokines intervenes in both acute and chronic GVHD should be better specified. It could be useful to specify it in the different tables (for example it could be modified by adding a column indicating the prevalent involvement in acute or chronic GVHD or both).

Respongse 2: Thank you for this important observation. We sincerely apologize for not specifying whether each interleukin primarily affects acute or chronic GVHD - this omission occurred because the cited literature frequently reports cytokine functions without clearly distinguishing between GVHD subtypes (particularly for IL-1/IL-6 families).We appreciate your patience and we have stepped up our review in this regard.

Comments 3: ROW 68: It is not indicated IL4 family, but it is indicated in the figure and in the text.

Respongse 3: Thank you for your careful review and for identifying this inconsistency. We have now added the IL-4 family to Row 69 as suggested, ensuring alignment with both the figure and text.

Comments 4: Table 4: Correct “TGF-b production and tissue fibrosisetc” in “TGF-b production and tissue fibrosis etc”

Respongse 4: Thank you for your careful review. We have corrected the formatting in Table 4 as suggested, revising "TGF-b production and tissue fibrosisetc" to "TGF-b production and tissue fibrosis etc."

Reviewer 3 Report

Comments and Suggestions for Authors

The present study about the “Interleukin Networks in GVHD: Mechanistic crosstalk, therapeutic targeting and emerging paradigms” is interesting.  However, authors should address the following questions

  • Authors are suggested to expand all the acronyms when they first appear in the manuscript
  • IL1, IL2, IL4, IL6, IL10, IL12, and IL17 …... family: The  Authors suggested including a small paragraph in all the cytokine families’ initiation with a group of cytokines involved and why they chose a few cytokines to explain in detail.
  • 1 Expand all the acronyms in the figure legend
  • What about the IL-4 cytokine, which was not included in either in IL-2 or IL-4 family?
  • What about the role of other group members of the IL6 family, like IL27, IL31, LIF, OIM etc.
  • What are the roles of other group members of the IL10 and IL17 families? Do the other cytokines not play a role in GVHD?

Author Response

Comments 1: Authors are suggested to expand all the acronyms when they first appear in the manuscript.

Response 1: Thank you for your helpful suggestion. We have now carefully reviewed the manuscript and ensured that all acronyms are fully expanded upon their first appearance in the text.

Comments 2: IL1, IL2, IL4, IL6, IL10, IL12, and IL17 …... family: The  Authors suggested including a small paragraph in all the cytokine families’ initiation with a group of cytokines involved and why they chose a few cytokines to explain in detail.

Response 2: Thank you for your constructive suggestion regarding the organization of cytokine family sections. We have now incorporated introductory paragraphs for each interleukin family at the beginning of their respective sections. These additions provide an overview of the full cytokine groups within each family while explaining our rationale for focusing on specific members that are most relevant to GVHD pathogenesis. The revised structure better contextualizes our selective coverage while maintaining the manuscript's depth on key cytokines, thereby improving the logical flow and reader comprehension.

Comments 3: 1 Expand all the acronyms in the figure legend

Response 3: Thank you for your helpful suggestion. We have now expanded all acronyms in the figure legends to their full terms as requested.

Comments 4: What about the IL-4 cytokine, which was not included in either in IL-2 or IL-4 family?

Response 4: Thank you for your valuable observation regarding the inclusion of IL-4 in our manuscript. We have carefully revised the content to incorporate a detailed analysis of IL-4's role in GVHD pathogenesis, ensuring its proper placement within the IL-4 family section alongside relevant mechanistic insights and supporting references. This addition has been thoughtfully integrated throughout the text and corresponding tables to maintain consistency and enhance the comprehensiveness of our cytokine family discussions.

Comments 5: What about the role of other group members of the IL6 family, like IL27, IL31, LIF, OIM etc.

Response 5: Thank you for your insightful suggestion regarding additional IL-6 family members. In response to your comment, we have now incorporated a detailed discussion of LIF within the IL-6 family section. Regarding other members you mentioned, while IL-27 has already been addressed in our IL-12 family discussion (as it belongs to both families), we conducted an extensive literature search but were unable to identify sufficient studies examining IL-31 or OSM (potentially Oncostatin M) in GVHD pathogenesis. We sincerely apologize for this limitation in our current coverage and would be grateful for any specific references you might recommend to strengthen this aspect of our review.

Comments 6: What are the roles of other group members of the IL10 and IL17 families? Do the other cytokines not play a role in GVHD?

Response 6: Thank you for raising this important question regarding other members of the IL-10 and IL-17 families. After conducting a thorough literature review, we found that research on most other cytokines within these families (beyond those we have discussed) in the context of GVHD remains quite limited in both preclinical and clinical studies. While we recognize these additional cytokines may potentially play roles in GVHD pathogenesis, the current evidence base appears insufficient to support substantive discussion in our review. We would be happy to incorporate any relevant studies you might suggest to address this gap more comprehensively in future revisions.

Round 2

Reviewer 1 Report

Comments and Suggestions for Authors

The manuscript entitled "Interleukin Networks in GVHD: Mechanistic Crosstalk, Therapeutic Targeting, and Emerging Paradigms" by Niu  et. al

is large text, which makes it harder to read through. The authors have to make a tough decision on unleashing it on readers. It points to the fact that authors do not have any idea on what area of GVHD and role of interleukins they want to focus on. It is hard to follow the article and the article fails to provide any directions on where the research for GVHD should take.

It is hard to approve this manuscript for publication since

  1. the authors are relying on a single reference for IL33. and 2 references for IL11.
  2. there response to the above comment is completely unrelated and is filled with typographical errors. see response to comment 6 in 1st review
  3. usually references in tables are listed as "References" but authors are using "Quote"
  4. There is a new section on LIF. This section is also written based on a single reference.
  5. The authors responded that the review is based on 80% animal studies and 20% human clinical trials. The authors should use this to divide and come up with two manuscripts. Role of interleukins in clinical trials of GVHD and mechanisms to reduce GVHD. 

Author Response

The manuscript entitled "Interleukin Networks in GVHD: Mechanistic Crosstalk, Therapeutic Targeting, and Emerging Paradigms" by Niu  et. al is large text, which makes it harder to read through. The authors have to make a tough decision on unleashing it on readers. It points to the fact that authors do not have any idea on what area of GVHD and role of interleukins they want to focus on. It is hard to follow the article and the article fails to provide any directions on where the research for GVHD should take. It is hard to approve this manuscript for publication since.

Respongse : We sincerely appreciate your constructive feedback, which has helped us significantly improve the focus and clarity of our manuscript. In response to your comments, we have thoroughly restructured the content to provide a more streamlined and reader-friendly discussion of interleukins in GVHD. The revised version now clearly distinguishes pro-inflammatory and anti-inflammatory cytokines, and separately discusses their roles in animal models and clinical experiments.We have removed redundant details, tightened the narrative flow, and ensured that each section contributes meaningfully to understanding how interleukin networks regulate GVHD pathogenesis and treatment.

Comments 1: the authors are relying on a single reference for IL33. and 2 references for IL11.

Respongse 1: We sincerely apologize for the oversight regarding the insufficient references for IL-33 and IL-11 in our original manuscript. We greatly appreciate the reviewer’s meticulous attention to this matter. As suggested, we have now expanded the discussion of both IL-33 and IL-11 in the revised manuscript, incorporating additional supporting literature to enhance our arguments.

Comments 2: there response to the above comment is completely unrelated and is filled with typographical errors. see response to comment 6 in 1st review.

Respongse 2: We sincerely apologize for the confusion and errors in our previous response regarding IL-13 and IL-11. We deeply appreciate the reviewer's patience in pointing out these mistakes. To thoroughly address this comment, we have now carefully revised the manuscript by significantly expanding the discussion of both IL-13 and IL-11, incorporating additional supporting references from both animal and clinical studies to provide a more comprehensive perspective. Specifically, we have added new experimental evidence from animal models demonstrating their mechanistic roles, along with clinical data highlighting their translational relevance. We are grateful for the opportunity to improve our work and would be happy to make any further modifications if needed.

Comments 3: usually references in tables are listed as "References" but authors are using "Quote"

Respongse 3: We sincerely appreciate the editor’s valuable suggestion regarding the formatting of references in tables. We agree that using the standard heading "References" instead of "Quote" enhances the professionalism and clarity of our manuscript. We have carefully revised all tables accordingly to ensure consistency with journal conventions. Thank you for this thoughtful guidance, which has undoubtedly improved the quality of our presentation. Please let us know if any additional adjustments are needed.

Comments 4: There is a new section on LIF. This section is also written based on a single reference.

Respongse 4: We sincerely appreciate the reviewer’s insightful comment regarding the section on LIF. In response, we have expanded the discussion by incorporating additional references to strengthen our coverage of this topic. However, we would like to note that research specifically investigating the role of LIF in GVHD remains relatively limited in the current literature. While we have included all relevant and available studies to support our discussion, we acknowledge that further investigation in this area would be valuable to fully elucidate LIF’s role in GVHD pathogenesis. We sincerely thank the reviewer for highlighting this point, and we hope our revised manuscript now provides a more balanced and comprehensive perspective on this topic. Please let us know if any further modifications would be helpful.

Comments 5: The authors responded that the review is based on 80% animal studies and 20% human clinical trials. The authors should use this to divide and come up with two manuscripts. Role of interleukins in clinical trials of GVHD and mechanisms to reduce GVHD. 

Respongse 5: We sincerely appreciate the editor’s valuable suggestion regarding the division of our manuscript into two distinct parts focusing on animal studies and clinical trials. We fully agree that this approach would enhance the clarity and impact of our research. In response, we have carefully restructured the content to first comprehensively examine interleukin mechanisms in GVHD pathogenesis using animal models, followed by a concise yet insightful summary of relevant clinical evidence in human studies. Once again, we are grateful for this constructive feedback, which has undoubtedly strengthened our work. Please let us know if any additional adjustments would be helpful.